# GeNIe: Generative Hard Negative Images Through Diffusion

## Abstract

Data augmentation is crucial in training deep models, preventing them from over-fitting to limited data. Recent advances in generative AI, e.g., diffusion models, have enabled more sophisticated augmentation techniques that produce data resembling natural images. We introduce `GeNIe` a novel augmentation method which leverages a latent diffusion model conditioned on a text prompt to combine two contrasting data points (an image from the source category and a text prompt from the target category) to generate challenging augmentations. To achieve this, we adjust the noise level (equivalently, number of diffusion iterations) to ensure the generated image retains low-level and background features from the source image while representing the target category, resulting in a *hard negative* sample for the source category. We further automate and enhance `GeNIe` by adaptively adjusting the noise level selection on a per image basis (coined as `GeNIe-Ada`), leading to further performance improvements. Our extensive experiments, in both few-shot and long-tail distribution settings, demonstrate the effectiveness of our novel augmentation method and its superior performance over the prior art.

## 1 Introduction

Augmentation has become an integral part of training deep learning models, particularly when faced with limited training data. For instance, when it comes to image classification with limited number of samples per class, model generalization ability can be significantly hindered. Simple transformations like rotation, cropping, and adjustments in brightness artificially diversify the training set, offering the model a more comprehensive grasp of potential data variations. Hence, augmentation can serve as a practical strategy to boost the model's learning capacity, minimizing the risk of overfitting and facilitating effective knowledge transfer from limited labelled data to real-world scenarios. Various image augmentation methods, encompassing standard transformations, and learning-based approaches have been proposed [16, 15, 110, 111, 100]. Some augmentation strategies combine two images possibly from two different categories to generate a new sample image. The simplest ones in this category are MixUp [111] and CutMix [110] where two images are combined in the pixel space. However, the resulting augmentations often do not lie within the manifold of natural images and act as out-of-distribution samples that will not be encountered during testing.

Recently, leveraging generative models for data augmentation has gained an upsurge of attention [100, 83, 63, 35]. These interesting studies, either based on fine-tuning or prompt engineering of diffusion models, are mostly focused on generating *generic augmentations* without considering the impact of other classes and incorporating that information into the generative process for a classification context. We take a different approach to generate challenging augmentations near the decision boundaries of a downstream classifier. Inspired by diffusion-based image editing methods [67, 63] some of which are previously used for data augmentation, we propose to use conditional latent dif-

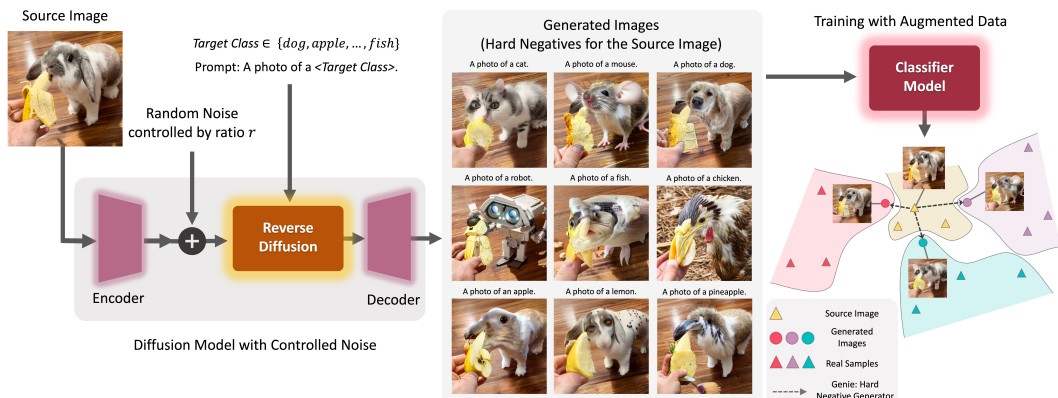

Figure 1: **Generative Hard Negative Images Through Diffusion (`GeNIe`):** generates hard negative images that belong to the target category but are similar to the source image from low-level feature and contextual perspectives. `GeNIe` starts from a source image passing it through a partial noise addition process, and conditioning it on a different target category. By controlling the amount of noise, the reverse latent diffusion process generates images that serve as *hard negatives* for the source category.

fusion models [81] for generating *hard negative* images. Our core idea (coined as `GeNIe`) is to sample source images from various categories and prompt the diffusion model with a contradictory text corresponding to a different target category. We demonstrate that the choice of noise level (or equivalently number of iterations) for the diffusion process plays a pivotal role in generating images that semantically belong to the target category while retaining low-level features from the source image. We argue that these generated samples serve as *hard negatives* [108, 65] for the source category (or from a dual perspective hard positives for the target category). To further enhance `GeNIe`, we propose an adaptive noise level selection strategy (dubbed as `GeNIe-Ada`) enabling it to adjust noise levels automatically per sample.

To establish the impact of `GeNIe`, we focus on two challenging scenarios: *long-tail* and *few-shot* settings. In real-world applications, data often follows a long-tail distribution, where common scenarios dominate and rare occurrences are underrepresented. For instance, a person jaywalking a highway causes models to struggle with such unusual scenarios. Combating such a bias or lack of sufficient data samples during model training is essential in building robust models for self-driving cars or surveillance systems, to name a few. Same challenge arises in few-shot learning settings where the model has to learn from only a handful of samples. Our extensive quantitative and qualitative experimentation, on a suite of few-shot and long-tail distribution settings, corroborate the effectiveness of the proposed novel augmentation method (`GeNIe`, `GeNIe-Ada`) in generating hard negatives, corroborating its significant impact on categories with a limited number of samples. A high-level sketch of `GeNIe` is illustrated in Fig. 1. Our main contributions are summarized below:

- We introduce `GeNIe`, a novel yet elegantly simple diffusion-based augmentation method to create challenging augmentations in the manifold of natural images. For the first time, to our best knowledge, `GeNIe` achieves this by combining two sources of information (a source image, and a contradictory target prompt) through a noise-level adjustment mechanism.

- We further extend `GeNIe` by automating the noise-level adjustment strategy on a per-sample basis (called `GeNIe-Ada`), to enable generating hard negative samples in the context of image classification, leading also to further performance enhancement.

- To substantiate the impact of `GeNIe`, we present a suit of quantitative and qualitative results including extensive experimentation on two challenging tasks: few-shot and long tail distribution settings corroborating that `GeNIe` (and its extension `GeNIe-Ada`) significantly improve the downstream classification performance.

## 2 Related Work

**Data Augmentations.** Simple flipping, cropping, colour jittering, and blurring are some forms of image augmentations [91]. These augmentations are commonly adopted in training deep learning models. However, using these data augmentations is not trivial in some domains. For example, using blurring might remove important low-level information from medical images. More advanced

approaches, such as MixUp [111] and CutMix [110], mix images and their labels accordingly [37, 59, 47, 17]. However, the resulting augmentations are not natural images anymore, and thus, act as out-of-distribution samples that will not be seen at test time. Another strand of research tailors the augmentation strategy through a learning process to fit the training data [23, 16, 15]. Unlike the above methods, we propose to utilize pre-trained latent diffusion models to generate hard negatives (in contrast to generic augmentations) through a noise adaptation strategy discussed in Section 3.

**Data Augmentation with Generative Models.** Using synthesized images from generative models to augment training data has been studied before in many domains [30, 86], including domain adaptation [41], visual alignment [71], and mitigation of dataset bias [88, 36, 73]. For example, [73] introduces a methodology aimed at enhancing test set evaluation through augmentation. While previous methods predominantly relied on GANs [114, 51, 101] as the generative model, more recent studies promote using diffusion models to augment the data [81, 35, 89, 100, 4, 62, 83, 42, 28, 26, 8]. More specifically, [100, 83, 35, 4] study the effectiveness of text-to-image diffusion models in data augmentation by diversification of each class with synthetic images. [100] leverages a text-to-image diffusion model and fine-tunes it on the downstream dataset using textual-inversion [31] to increase the diversity of existing samples. [83] also utilizes a text-to-image diffusion model, but with a BLIP [53] model to generate meaningful captions from the existing images. [42] utilizes diffusion models for augmentation to correct model mistakes. [28] uses CLIP [76] to filter generated images. [26] utilizes text-based diffusion and a large language model (LLM) to diversify the training data. [8] uses an LLM to generate text descriptions of failure modes associated with spurious correlations, which are then used to generate synthetic data through generative models. The challenge here is that the LLM has little understanding of such failure scenarios and contexts.

We take a completely different approach here, without replying on any extra source of information (e.g., through an LLM). Inspired by image editing approaches such as Boomerang [63] and SDEdit [67], we propose to adaptively guide a latent diffusion model to generate *hard negatives* images [65, 108] on a per-sample basis per category. In a nutshell, the aforementioned studies focus on improving the diversity of each class with effective prompts and diffusion models, however, we focus on generating effective *hard negative* samples for each class by combining two sources of contradicting information (images from the source category and text prompt from the target category).

**Language Guided Recognition Models.** Vision-Language foundation models (VLMs) [2, 76, 81, 84, 77, 78] utilize human language to guide the generation of images or to extract features from images that are aligned with human language. For example, CLIP [76] shows decent zero-shot performance on many downstream tasks by matching images to their text descriptions. Some recent works improve the utilization of human language in the prompt [25, 72], and others use a diffusion model directly as a classifier [49]. Similar to the above, we use a foundation model (Stable Diffusion 1.5 [81]) to improve the downstream task. Concretely, we utilize category names of the downstream tasks to augment their associate training data with hard negative samples.

**Few-Shot Learning.** In Few-shot Learning (FSL), we pre-train a model with abundant data to learn a rich representation, then fine-tune it on new tasks with only a few available samples. In supervised FSL [10, 1, 74, 109, 27, 54, 95, 116, 92], pretraining is done on a labeled dataset, whereas in unsupervised FSL [43, 103, 61, 75, 3, 46, 39, 66, 90] the pre-training has to be conducted on an unlabeled dataset. We assess the impact of `GeNIe` on a number of few-shot scenarios and state-of-the-art baselines by accentuating on its impact on the few-shot inference stage.

# 3 Proposed Method: `GeNIe`

Given a source image $X_S$ from category S = <source category>, we are interested in generating a target image $X_r$ from category $T = $ <target category>. In doing so, we intend to ensure the low-level visual features or background context of the source image are preserved, so that we generate samples that would serve as *hard negatives* for the *source* image. To this aim, we adopt a conditional latent diffusion model (such as Stable Diffusion, [81]) conditioned on a text prompt of the following format "A photo of a $T = $ <target category>".

**Key Idea.** `GeNIe` in its basic form is a simple yet effective augmentation sample generator for improving a classifier $f_\theta(.)$ with the following two key aspects: (i) inspired by [63, 67] instead of adding the full amount of noise $\sigma_{max}$ and going through all $N_{max}$ (being typically 50) steps of denoising, we use less amount of noise ($r\sigma_{max}$, with $r \in (0, 1)$) and consequently fewer number of denoising iterations ($\lfloor rN_{max} \rfloor$); (ii) we prompt the diffusion model with a $P$ mandating a target

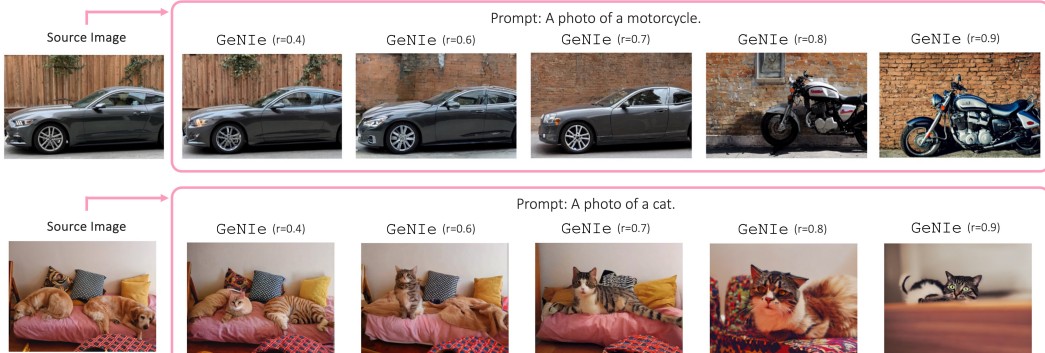

Figure 2: **Effect of noise ratio, $r$, in `GeNIe`:** we employ `GeNIe` to generate augmentations for the target classes (motorcycle and cat) with varying $r$. Smaller $r$ yields images closely resembling the source semantics, creating an inconsistency with the intended target label. By tracing $r$ from 0 to 1, augmentations gradually transition from source image characteristics to the target category. However, a distinct shift from the source to the target occurs at a specific $r$ that may vary for different source images or target categories. For more examples, please refer to Fig. A4.

category $T$ different than the source $S$. Hence, we denote the conditional diffusion process as $X_r = \texttt{STDiff}(X_S, P, r)$. In such a construct, the proximity of the final decoded image $X_r$ to the source image $X_S$ or the target category defined through the text prompt $P$ depends on $r$. Hence, by controlling the amount of noise, we can generate images that blend characteristics of both the text prompt $P$ and the source image $X_S$. If we do not provide much of visual details in the text prompt (e.g., desired background, etc.), we expect the decoded image $X_r$ to follow the details of $X_S$ while reflecting the semantics of the text prompt $P$. We argue, and demonstrate later, that the newly generated samples can serve as *hard negative* examples for the source category $S$ since they share the low-level features of $X_S$ while representing the semantics of the target category, $T$. Notably, the source category $S$ can be randomly sampled or be carefully extracted from the confusion matrix of $f_\theta(.)$ based on real training data. The latter might result in even *harder negative* samples being now cognizant of model confusions. Finally, we will append our initial dataset with the newly generated hard negative samples through `GeNIe` and (re)train the classifier model.

**Enhancing `GeNIe`: `GeNIe-Ada`.** One of the remarkable aspects of `GeNIe` lies in its simple application, requiring only $X_S$, $P$, and $r$. However, selecting the appropriate value for $r$ poses a challenge as it profoundly influences the outcome. When $r$ is small, the resulting $X_r$ tends to closely resemble $X_S$, and conversely, when $r$ is large (closer to 1), it tends to resemble the semantics of the target category. This phenomenon arises because a smaller noise level restricts the capacity of the diffusion model to deviate from the semantics of the input $X_S$. Thus, a critical question emerges: how can we select $r$ for a particular source image to generate samples that preserve the low-level semantics of the source category $S$ in $X_S$ while effectively representing the semantics of the target category $T$? We propose a method to determine an ideal value for $r$.

Our intuition suggests that by varying the noise ratio $r$ from 0 to 1, $X_r$ will progressively resemble category $S$ in the beginning and category $T$ towards the end. However, somewhere between 0 and 1, $X_r$ will undergo a rapid transition from category $S$ to $T$. This phenomenon is empirically observed in our experiments with varying $r$, as depicted in Fig. 2. Although the exact reason for this rapid change remains uncertain, one possible explanation is that the intermediate points between two categories reside far from the natural image manifold, thus, challenging the diffusion model's capability to generate them. Ideally, we should select $r$ corresponding to just after this rapid semantic transition, as at this point, $X_r$ exhibits the highest similarity to the source image while belonging to the target category.

We propose to trace the semantic trajectory between $X_S$ and $X_T$ through the lens of the classifier $f_\theta(.)$. As shown in Algorithm 1, assuming access to the classifier backbone $f_\theta(.)$ and at least one example $X_T$ from the target category, we convert both $X_S$ and $X_T$ into their respective latent vectors $Z_S$ and $Z_T$ by passing them through $f_\theta(.)$. Then, we sample $M$ values for $r$ uniformly distributed $\in (0, 1)$, generating their corresponding $X_r$ and their latent vectors $Z_r$ for all those $r$. Subsequently, we calculate $d_r = \frac{(Z_r - Z_S)^T (Z_T - Z_S)}{||Z_T - Z_S||_2}$ as the distance between $Z_r$ and $Z_S$ projected onto the vector connecting $Z_S$ and $Z_T$. Our hypothesis posits that the rapid semantic transition corresponds to a sharp change in this projected distance. Therefore, we sample $n$ values for $r$ uniformly distributed

**Algorithm 1:** `GeNIe-Ada`

**Require:** $X_S, X_T, f_\theta(.), \texttt{STDiff}(.), M$
Extract $Z_S \leftarrow f_\theta(X_s), Z_T \leftarrow f_\theta(X_T)$
**for** $m \in [1, M]$ **do**
$\quad r \leftarrow \frac{m}{M}, Z_r \leftarrow f_\theta(\,\texttt{STDiff}(X, P, r)\,)$
$\quad d_m \leftarrow \frac{(Z_r - Z_S)^T (Z_T - Z_S)}{||Z_T - Z_S||_2}$
$m^* \leftarrow \operatorname{argmax}_m |d_m - d_{m-1}|, \forall m \in [2, M]$
$r^* \leftarrow \frac{m^*}{n}$
**Return:** $X_{r^*} = \texttt{STDiff}(X_S, P, r^*)$

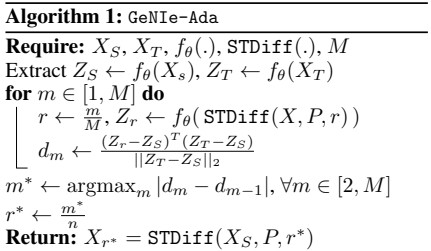
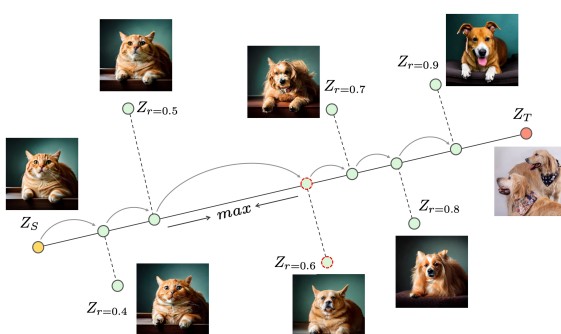

Figure 3: `GeNIe-Ada`: To choose $r$ adaptively for each (source image, target category) pair, we propose tracing the semantic trajectory from $Z_S$ (source image embeddings) to $Z_T$ (target embeddings) through the lens of the classifier $f_\theta(\cdot)$ (Algorithm 1). We adaptively select the sample right after the largest semantic shift.

between 0 and 1, and analyze the variations in $d_r$. We identify the largest gap in $d_r$ and select the $r$ value just after the gap when increasing $r$, as detailed in Algorithm 1 and illustrated in Fig. 3.

# 4 Experiments

Since the impact of augmentation is more pronounced when the training data is limited, we evaluate the impact of `GeNIe` on Few-Shot classification in Section 4.1, Long-Tailed classification in Section 4.2, and fine-grained classification in Section A.2. For `GeNIe-Ada` in all scenarios, we utilize `GeNIe` to generate augmentations from the noise level set $\{0.5, 0.6, 0.7, 0.8, 0.9\}$. The selection of the appropriate noise level per source image and target is adaptive, achieved through Algorithm 1.

**Baselines.** We use Stable Diffusion 1.5 [81] as our base diffusion model. In all settings, we use the same prompt format to generate images for the target class: i.e., "A photo of a $<$target category$>$", where we replace the `target category` with the target category label. We generate $512 \times 512$ images for all methods. For fairness in comparison, we generate the same number of new images for each class. We use a single NVIDIA RTX 3090 for image generation. We consider 4 diffusion-based baselines and a suite of traditional data augmentation baselines:

`Img2Img` [63, 67]: We sample an image from a target class, add noise to its latent representation and then pass it along with a prompt for the target category through reverse diffusion. The focus here is on a target class for which we generate extra positive samples. Adding large amount of noise leads to generating an image less similar to the original image. We use two different noise magnitudes for this baseline: $r = 0.3$ and $r = 0.7$ and denote them by $\texttt{Img2Img}^L$ and $\texttt{Img2Img}^H$, respectively.

`Txt2Img` [4, 35]: For this baseline, we omit the forward diffusion process and only use the reverse process starting from a text prompt for the target class of interest. This is similar to the base text-to-image generation strategy adopted in [81, 35, 89, 4, 62]. Fig. 4 illustrates a set of generated augmentation examples for `Txt2Img`, `Img2Img`, and `GeNIe`.

DAFusion [100]: In this method, an embedding is optimized with a set of images for each class to correspond to the classes in the dataset. This approach is introduced in Textual Inversion [32]. We optimize an embedding for 5000 iterations for each class in the dataset, followed by augmentation similar as the DAFusion method.

Cap2Aug[83]: It is a recent diffusion-based data augmentation strategy that uses image captions as text prompts for an image-to-image diffusion model.

**Traditional Data Augmentation:** We consider both weak and strong traditional augmentations. More specifically, for weak augmentation we use random resize crop with scaling $\in [0.2, 1.0]$ and horizontal flipping. For strong augmentation, we consider random color jitter, random grayscale, and Gaussian blur. For the sake of completeness, we also compare against data augmentations such as CutMix [110] and MixUp [111] that combine two images together.

## 4.1 Few-shot Classification

We assess the impact of `GeNIe` compared to other augmentations in a number of few-shot classification (FSL) scenarios, where the model has to learn only from the samples contained in the ($N$-way, $K$-shot) support set and infer on the query set. Note that this corresponds to an inference-only FSL

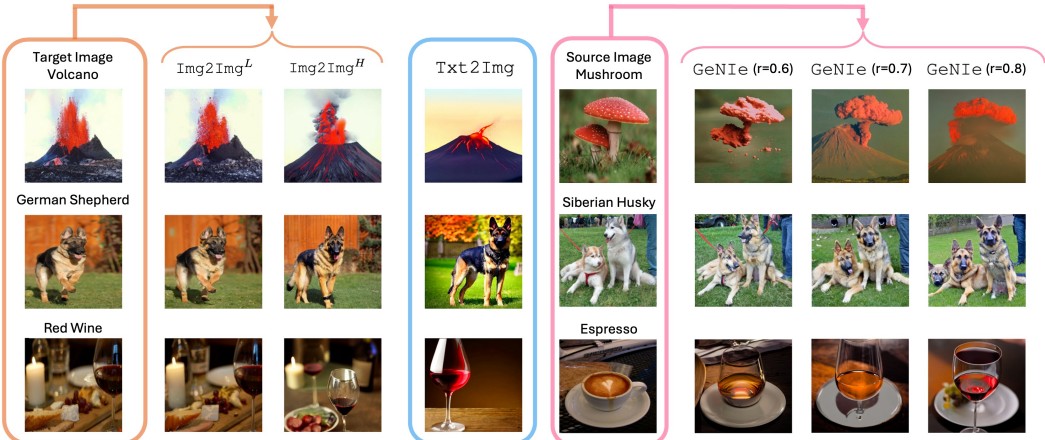

Figure 4: **Visualization of Generative Samples:** We compare `GeNIe` with two baselines: `Img2Img`$^L$ **augmentation:** both image and text prompt are from the same category. Adding noise does not change the image much, so they are not hard examples. `Txt2Img` **augmentation:** We simply use the text prompt only to generate an image for the desired category (e.g., using a text2image method). Such images may be far from the domain of our task since the generation is not informed by any visual data from our task. `GeNIe` **augmentation:** We use the target category name in the text prompt only along with the source image.

setting where a pretraining stage on an abundant dataset is discarded. The goal is to assess how well the model can benefit from the augmentations while keeping the original $N \times K$ samples intact.

**Datasets.** We conduct our few-shot experiments on two most commonly adopted few-shot classification datasets: *mini*-Imagenet [79] and *tiered*-Imagenet [80]. *mini*-Imagenet is a subset of ImageNet [22] for few-shot classification. It contains 100 classes with 600 samples each. We follow the predominantly adopted settings of [79, 10] where we split the entire dataset into 64 classes for training, 16 for validation and 20 for testing. *tiered*-Imagenet is a larger subset of ImageNet with 608 classes and a total of 779, 165 images, which are grouped into 34 higher-level nodes in the *ImageNet* human-curated hierarchy. This set of nodes is partitioned into 20, 6, and 8 disjoint sets of training, validation, and testing nodes, and the corresponding classes form the respective meta-sets.

**Evaluation.** To quantify the impact of different augmentation methods, we evaluate the test-set accuracies of a state-of-the-art unsupervised few-shot learning method with `GeNIe` and compare them against the accuracies obtained using other augmentation methods. Specifically, we use UniSiam [61] pre-trained with ResNet-18, ResNet-34 and ResNet-50 backbones and follow its evaluation strategy of fine-tuning a logistic regressor to perform ($N$-way, $K$-shot) classification on the test sets of *mini*- and *tiered*-Imagenet. Following [79], an episode consists of a labeled support-set and an unlabelled query-set. The support-set contains $N$ randomly sampled classes where each class contains $K$ samples, whereas the query-set contains $Q$ randomly sampled unlabeled images per class. We conduct our experiments on the two most commonly adopted settings: (5-way, 1-shot) and (5-way, 5-shot) classification settings. Following the literature, we sample 16-shots per class for the query set in both settings. We report the test accuracies along with the 95% confidence interval over 600 and 1000 episodes for *mini*-ImageNet and *tiered*-ImageNet, respectively.

**Implementation Details:** `GeNIe` generates augmented images for each class using images from all other classes as the source image. We use $r = 0.8$ in our experiments. We generate 4 samples per class as augmentations in the 5-way, 1-shot setting and 20 samples per class as augmentations in the 5-way, 5-shot setting. For the sake of a fair comparison, we ensure that the total number of labelled samples in the support set after augmentation remains the same across all different traditional and generative augmentation methodologies. Due to the expensive training of embeddings for each class in each episode, we only evaluated the DA-Fusion baseline on the first 100 episodes.

**Results:** The results on *mini*-Imagenet and *tiered*-Imagenet for both (5-way, 1 and 5-shot) settings are summarized in Table 1 and Table 2, respectively. Regardless of the choice of backbone, we observe that `GeNIe` helps consistently improve UniSiam's performance and outperform other supervised and unsupervised few-shot classification methods as well as other diffusion-based [100, 63, 82, 35] and classical [110, 111] data augmentation techniques on both datasets, across both (5-way, 1 and 5-shot) settings. Our noise adaptive method of selecting optimal augmentations per source image (GeNIe-Ada) further improves `GeNIe`'s performance across all three backbones, both

Table 1: ***mini*-ImageNet:** We use our augmentations on (5-way, 1-shot) and (5-way, 5-shot) few-shot settings of mini-Imagenet dataset with 3 different backbones (ResNet-18, 34, and 50). We compare with various baselines and show that our augmentations with UniSiam outperform all the baselines including `Txt2Img` and DAFusion augmentation. The number of generated images per class is 4 for 1-shot and 20 for 5-shot settings.

**ResNet-18**

| Augmentation | Method | Pre-training | 1-shot | 5-shot |
|---|---|---|---|---|
| - | iDeMe-Net [14] | sup. | 59.1±0.9 | 74.6±0.7 |
| - | Robust + dist [27] | sup. | 63.7±0.6 | 81.2±0.4 |
| - | AFHN [54] | sup. | 62.4±0.7 | 78.2±0.6 |
| Weak | ProtoNet+SSL [94] | sup.+ssl | - | 76.6 |
| Weak | Neg-Cosine [57] | sup. | 62.3±0.8 | 80.9±0.6 |
| - | Centroid Align[1] | sup. | 59.9±0.7 | 80.4±0.7 |
| - | Baseline [10] | sup. | 59.6±0.8 | 77.3±0.6 |
| - | Baseline++ [10] | sup. | 59.0±0.8 | 76.7±0.6 |
| Weak | PSST [13] | sup.+ssl | 59.5±0.5 | 77.4±0.5 |
| Weak | UMTRA [46] | unsup. | 43.1±0.4 | 53.4±0.3 |
| Weak | ProtoCLR [66] | unsup. | 50.9±0.4 | 71.6±0.3 |
| Weak | SimCLR [9] | unsup. | 62.6±0.4 | 79.7±0.3 |
| Weak | SimSiam [12] | unsup. | 62.8±0.4 | 79.9±0.3 |
| Weak | UniSiam+dist [61] | unsup. | **64.1±0.4** | **82.3±0.3** |
| Weak | UniSiam [61] | unsup. | 63.1±0.8 | 81.4±0.5 |
| Strong | UniSiam [61] | unsup. | 62.8±0.8 | 81.2±0.6 |
| CutMix [110] | UniSiam [61] | unsup. | 62.7±0.8 | 80.6±0.6 |
| MixUp [111] | UniSiam [61] | unsup. | 62.1±0.8 | 80.7±0.6 |
| Img2Img$^L$[63] | UniSiam [61] | unsup. | 63.9±0.8 | 82.1±0.5 |
| Img2Img$^H$[63] | UniSiam [61] | unsup. | 69.1±0.7 | 84.0±0.5 |
| Txt2Img[4, 35] | UniSiam [61] | unsup. | 74.1±0.6 | 84.6±0.5 |
| DAFusion [100] | UniSiam [61] | unsup. | 64.3±1.8 | 82.0±1.4 |
| GeNIe (Ours) | UniSiam [61] | unsup. | **75.5±0.6** | **85.4±0.4** |
| GeNIe-Ada (Ours) | UniSiam [61] | unsup. | **76.8±0.6** | **85.9±0.4** |

**ResNet-34**

| Augmentation | Method | Pre-training | 1-shot | 5-shot |
|---|---|---|---|---|
| Weak | Baseline [10] | sup. | 49.8±0.7 | 73.5±0.7 |
| Weak | Baseline++ [10] | sup. | 52.7±0.8 | 76.2±0.6 |
| Weak | SimCLR [9] | unsup. | 64.0±0.4 | 79.8±0.3 |
| Weak | SimSiam [12] | unsup. | 63.8±0.4 | 80.4±0.3 |
| Weak | UniSiam+dist [61] | unsup. | **65.6±0.4** | **83.4±0.2** |
| Weak | UniSiam [61] | unsup. | 64.3±0.8 | 82.3±0.5 |
| Strong | UniSiam [61] | unsup. | 64.5±0.8 | 82.1±0.6 |
| CutMix [110] | UniSiam [61] | unsup. | 64.0±0.8 | 81.7±0.6 |
| MixUp [111] | UniSiam [61] | unsup. | 63.7±0.8 | 80.1±0.8 |
| Img2Img$^L$[63] | UniSiam [61] | unsup. | 65.5±0.8 | 82.9±0.5 |
| Img2Img$^H$[63] | UniSiam [61] | unsup. | 70.5±0.8 | 84.8±0.5 |
| Txt2Img[4, 35] | UniSiam [61] | unsup. | 75.4±0.6 | 85.5±0.5 |
| DAFusion [100] | UniSiam [61] | unsup. | 64.7±1.9 | 83.2±1.4 |
| GeNIe (Ours) | UniSiam [61] | unsup. | **77.1±0.6** | **86.3±0.4** |
| GeNIe-Ada (Ours) | UniSiam [61] | unsup. | **78.5±0.6** | **86.6±0.4** |

**ResNet-50**

| Augmentation | Method | Pre-training | 1-shot | 5-shot |
|---|---|---|---|---|
| Weak | PDA+Net [11] | unsup. | 63.8±0.9 | 83.1±0.6 |
| Weak | Meta-DM [40] | unsup. | 66.7±0.4 | 85.3±0.2 |
| Weak | UniSiam [61] | unsup. | 64.6±0.8 | 83.4±0.5 |
| Strong | UniSiam [61] | unsup. | 64.8±0.8 | 83.2±0.5 |
| CutMix [110] | UniSiam [61] | unsup. | 64.3±0.8 | 83.2±0.5 |
| MixUp [111] | UniSiam [61] | unsup. | 63.8±0.8 | 84.6±0.5 |
| Img2Img$^L$[63] | UniSiam [61] | unsup. | 66.0±0.8 | 84.0±0.5 |
| Img2Img$^H$[63] | UniSiam [61] | unsup. | 71.1±0.7 | 85.7±0.5 |
| Txt2Img[4, 35] | UniSiam [61] | unsup. | 76.4±0.6 | 86.5±0.4 |
| DAFusion [100] | UniSiam [61] | unsup. | 65.7±1.8 | 83.9±1.2 |
| GeNIe (Ours) | UniSiam [61] | unsup. | **77.3±0.6** | **87.2±0.4** |
| GeNIe-Ada (Ours) | UniSiam [61] | unsup. | **78.6±0.6** | **87.9±0.4** |

few-shot settings, and both datasets (*mini* and *tiered*-Imagenet). Few-shot accuracies for ResNet-34 computed on *tiered*Imagenet are reported in Section A.3 of the appendix. Note that employing CutMix and MixUp seems to lead to performance degradation compared to weak augmentations, probably due to overfitting since these methods can only choose from 4 other classes to mix.

## 4.2 Long-Tailed Classification

We evaluate our method on long-tailed data, where the number of instances per class is unbalanced, with most categories having limited samples (tail). Our goal is to mitigate this bias by augmenting the tail of the distribution with generated samples. We evaluate `GeNIe` using two different backbones and methods: the ViT architecture with LViT [107], and ResNet50 with VL-LTR [97].

Following LViT [107], we first train an MAE [34] and ViT on the unbalanced dataset without any augmentation. Next, we train the Balanced Fine-Tuning stage of LViT by incorporating the augmentation data generated using `GeNIe` or other baselines. For ResNet50, we use VL-LTR code to fine-tune the CLIP [76] ResNet50 pretrained backbone with generated augmentations by `GeNIe`.

**Dataset:** We perform experiments on ImageNet-LT [60]. It contains 115.8K images from $1,000$ categories. The number of images per class varies from 1280 to 5. Imagenet-LT classes can be divided into 3 groups: "Few" with less than 20 images, "Med" with $20 - 100$ images, and "Many" with more than 100 images. Imagenet-LT uses the same validation set as ImageNet. We augment "Few" categories only and limit the number of generated images to 50 samples per class. For `GeNIe`, instead of randomly sampling the source images from other classes, we use a confusion matrix on the training data to find the top-4 most confused classes and only consider those classes for random sampling of the source image. The source category may be from "Many", "Med", or "Few sets".

**Results:** Augmenting training data with `GeNIe-Ada` improves accuracy on the "Few" set by $11.7\%$ and $4.4\%$ compared with LViT only and LViT with `Txt2Img` augmentation baselines respectively. In ResNet50, `GeNIe-Ada` outperforms Cap2Aug baseline in "Few" categories by $7.6\%$. The results are summarized in Table 3. Please refer to Section A.4 for implementation details.

## 4.3 Ablation and Analysis

**Semantic Shift from Source to Target Class.** The core motivation behind `GeNIe-Ada` is that by varying the noise ratio $r$ from 0 to 1, augmented sample $X_r$ will progressively shift its semantic category from source ($S$) in the beginning to target category ($T$) towards the end. However, somewhere between 0 and 1, $X_r$ will undergo a rapid transition from $S$ to $T$. To demonstrate this hypothesis empirically, in Figs. 5 and A5, we visualize pairs of source images and target categories with their respective `GeNIe` generated augmentations for different noise ratios $r$, along with their corresponding

Table 2: *tiered*-**ImageNet:** Accuracies (% ± std) for 5-way, 1-shot and 5-way, 5-shot classification settings on the test-set. We compare against various SOTA supervised and unsupervised few-shot classification baselines as well as other augmentation methods, with UniSiam [61] pre-trained ResNet-18,50 backbones.

| Augmentation | Method | Pre-training | 1-shot | 5-shot |
|---|---|---|---|---|
| **ResNet-18** | | | | |
| Weak | SimCLR[9] | unsup. | 63.4±0.4 | 79.2±0.3 |
| Weak | SimSiam [12] | unsup. | 64.1±0.4 | 81.4±0.3 |
| Weak | UniSiam [61] | unsup. | 63.1±0.7 | 81.0±0.5 |
| Strong | UniSiam [61] | unsup. | 62.8±0.7 | 80.9±0.5 |
| CutMix [110] | UniSiam [61] | unsup. | 62.1±0.7 | 78.9±0.6 |
| MixUp [111] | UniSiam [61] | unsup. | 62.1±0.7 | 78.4±0.6 |
| Img2Img$^L$[63] | UniSiam [61] | unsup. | 63.9±0.7 | 81.8±0.5 |
| Img2Img$^H$[63] | UniSiam [61] | unsup. | 68.7±0.7 | 83.5±0.5 |
| Txt2Img[35] | UniSiam [61] | unsup. | 72.9±0.6 | 84.2±0.5 |
| DAFusion [100] | UniSiam [61] | unsup. | 62.6±2.1 | 81.0±1.5 |
| GeNIe(Ours) | UniSiam [61] | unsup. | 73.6±0.6 | 85.0±0.4 |
| GeNIe-Ada(Ours) | UniSiam [61] | unsup. | 75.1±0.6 | 85.5±0.5 |
| **ResNet-50** | | | | |
| Weak | PDA+Net [11] | unsup. | 69.0±0.9 | 84.2±0.7 |
| Weak | Meta-DM [40] | unsup. | 69.6±0.4 | 86.5±0.3 |
| Weak | UniSiam + dist [61] | unsup. | 69.6±0.4 | 86.5±0.4 |
| Weak | UniSiam [61] | unsup. | 66.8±0.7 | 84.7±0.5 |
| Strong | UniSiam [61] | unsup. | 66.5±0.7 | 84.5±0.5 |
| CutMix [110] | UniSiam [61] | unsup. | 66.0±0.7 | 83.3±0.5 |
| MixUp [111] | UniSiam [61] | unsup. | 66.1±0.5 | 84.1±0.8 |
| Img2Img$^L$[63] | UniSiam [61] | unsup. | 67.8±0.7 | 85.3±0.5 |
| Img2Img$^H$[63] | UniSiam [61] | unsup. | 72.4±0.7 | 86.7±0.4 |
| Txt2Img[35] | UniSiam [61] | unsup. | 77.1±0.6 | 87.3±0.4 |
| DAFusion [100] | UniSiam [61] | unsup. | 66.5±2.2 | 84.8±1.4 |
| GeNIe (Ours) | UniSiam [61] | unsup. | 78.0±0.6 | 88.0±0.4 |
| GeNIe-Ada (Ours) | UniSiam [61] | unsup. | 78.8±0.6 | 88.6±0.6 |

Table 3: **Long-Tailed ImageNet-LT:** We compare different augmentation methods on ImageNet-LT and report Top-1 accuracy for "Few", "Medium", and "Many" sets. On the "Few" set and LiVT method, our augmentations improve the accuracy by 11.7 points compared to LiVT original augmentation and 4.4 points compared to Txt2Img. GeNIe-Ada outperforms Cap2Aug baseline in "Few" categories by 7.6%. Refer to Table A4 for a full comparison with prior Long-Tailed methods.

| Method | Many | Med. | Few | Overall Acc |
|---|---|---|---|---|
| **ResNet-50** | | | | |
| ResLT [18] | 63.3 | 53.3 | 40.3 | 55.1 |
| PaCo [19] | 68.2 | 58.7 | 41.0 | 60.0 |
| LWS [44] | 62.2 | 48.6 | 31.8 | 51.5 |
| Zero-shot CLIP [76] | 60.8 | 59.3 | 58.6 | 59.8 |
| DRO-LT [85] | 64.0 | 49.8 | 33.1 | 53.5 |
| VL-LTR [97] | 77.8 | 67.0 | 50.8 | 70.1 |
| Cap2Aug [83] | 78.5 | **67.7** | 51.9 | 70.9 |
| GeNIe-Ada | **79.2** | 64.6 | **59.5** | **71.5** |
| **ViT-B** | | | | |
| Method | Many | Med. | Few | Overall Acc |
| ViT [24] | 50.5 | 23.5 | 6.9 | 31.6 |
| MAE [33] | 74.7 | 48.2 | 19.4 | 54.5 |
| DeiT [99] | 70.4 | 40.9 | 12.8 | 48.4 |
| LiVT [107] | 73.6 | 56.4 | 41.0 | 60.9 |
| LiVT + Img2Img$^L$ | 74.3 | 56.4 | 34.3 | 60.5 |
| LiVT + Img2Img$^H$ | 73.8 | 56.4 | 45.3 | 61.6 |
| LiVT + Txt2Img | **74.9** | 55.6 | 48.3 | 62.2 |
| LiVT + GeNIe-Ada | 74.0 | **56.9** | **52.7** | **63.1** |

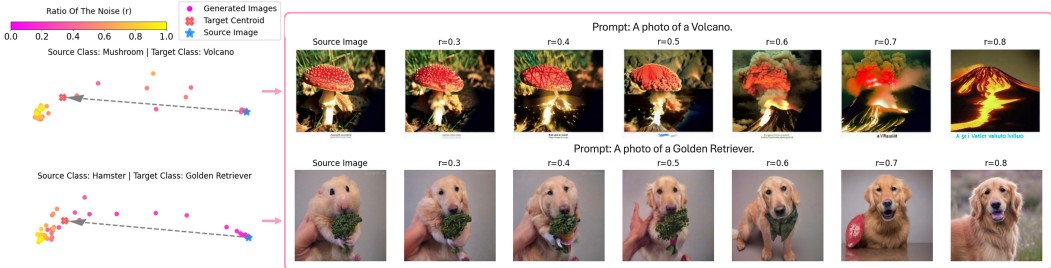

Figure 5: **Embedding visualizations of generative augmentations:** We pass all generative augmentations through DINOv2 ViT-G (serving as an oracle) to extract their corresponding embeddings and visualize them with PCA. As shown, the extent of semantic shifts varies based on both the source image and the target class.

PCA-projected embedding scatter plots (on the far left). We extract embeddings for all the images using a DINOv2 ViT-G pretrained backbone, which we assume as an oracle model in identifying the right category. We observe that as $r$ increases from $0.3$ to $0.8$, the images transition to embody more of the target category's semantics while preserving the contextual features of the source image. This transition of semantics can also be observed in the embedding plots (on the left) where they consistently shift from the proximity of the source image (blue star) to the target class's centroid (red cross) as the noise ratio $r$ increases. The sparse distribution of points within $r = [0.4, 0.6]$ for the first image and $r = [0.2, 0.4]$ for the second image aligns with our intuition of a rapid transition from category $S$ to $T$, thus empirically affirming our motivation behind GeNIe-Ada.

To further establish this, in Fig. 6, we demonstrate the efficacy of GeNIe in generating hard negatives at the decision boundaries of an SVM classifier, which is trained on the labelled support set of the few-shot tasks of *mini*-Imagenet, without any augmentations. We then plot source and target class probabilities ($P(Y_S|X_r)$ and $P(Y_T|X_r)$, respectively) of the generated augmentation samples $X_r$. For both $r = 0.6$ and $0.7$, there is significant overlap between $P(Y_S|X_r)$ and $P(Y_T|X_r)$, making it difficult for the classifier to decide the correct class. On the right-hand-side, GeNIe-Ada automatically selects the best $r$ resulting in the most overlap between the two distributions, thus offering the hardest negative sample among the considered $r$ values (for more details see A.1). Note that a large overlap between distributions is not sufficient to call the generated samples hard negatives because they should also belong to the target category. This is, however, confirmed by the high Oracle accuracy in Table 4 (elaborated in detail in the following paragraph) which verifies that majority of the generated augmentation samples do belong to the target category.

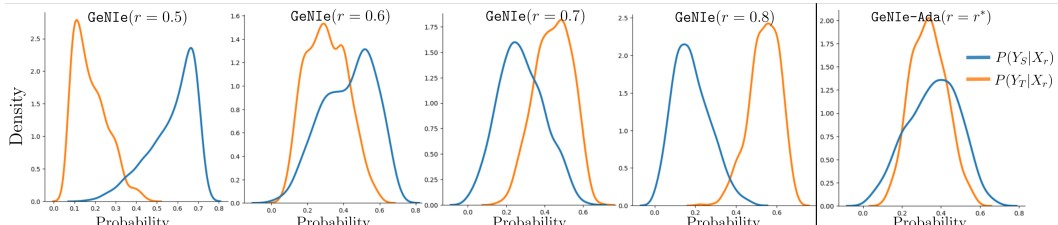

Figure 6: **Why `GeNIe` augmentations are challenging?** While deciding which class the generated augmentations ($X_r$) belong to is already difficult within $r = [0.6, 0.7]$ (due to high overlap between $P(Y_S|X_r)$ and $P(Y_T|X_r)$), `GeNIe-Ada` selects the best noise threshold ($r^*$) offering the hardest negative sample.

Table 4: **Effect of Noise in `GeNIe`:** We use the same setting as in Table 1 to study the effect of the amount of noise. As expected (also shown in Fig 5), small noise results in worse accuracy since some generated images may be from the source category rather than the target one. For $r = 0.5$ only 73% of the generated data is from the target category. This behaviour is also shown in Fig. 2. Notably, reducing the noise level below 0.7 is associated with a decline in oracle accuracy and subsequent degradation in the performance of the final few-shot model. Note that the high oracle accuracy of `GeNIe-Ada` demonstrates its capability to adaptively select the noise level per source and target, ensuring semantic consistency with the intended target.

| Noise | ResNet-18 | | ResNet-34 | | ResNet-50 | | Oracle |
| | 1-shot | 5-shot | 1-shot | 5-shot | 1-shot | 5-shot | Acc |
|---|---|---|---|---|---|---|---|
| GeNIe(r=0.5) | 60.42±0.8 | 74.11±0.6 | 62.02±0.8 | 75.80±0.6 | 63.65±0.9 | 77.61±0.6 | 73.4±0.5 |
| GeNIe(r=0.6) | 69.66±0.7 | 80.65±0.5 | 71.13±0.7 | 82.21±0.5 | 72.10±0.7 | 82.79±0.5 | 85.8±0.4 |
| GeNIe(r=0.7) | 74.50±0.6 | 83.26±0.5 | 76.41±0.6 | 84.44±0.5 | 77.05±0.6 | 84.95±0.4 | 94.5±0.2 |
| GeNIe(r=0.8) | 75.45±0.6 | 85.38±0.4 | 77.08±0.6 | 86.28±0.4 | 77.28±0.6 | 87.22±0.4 | 98.2±0.1 |
| GeNIe(r=0.9) | 74.96±0.6 | 85.29±0.4 | 77.63±0.6 | 86.17±0.4 | 77.73±0.6 | 87.00±0.4 | 99.3±0.1 |
| GeNIe-Ada | 76.79±0.6 | 85.89±0.4 | 78.49±0.6 | 86.55±0.4 | 78.64±0.6 | 87.88±0.4 | 98.9±0.2 |

**Label consistency of the generated samples.** The choice of noise ratio $r$ is important in producing hard negative examples. In Table 4, we present the accuracy of the `GeNIe` model across various noise ratios, alongside the oracle accuracy, which is an ImageNet pre-trained DeiT-Base [98] classifier. We observe a decline in the label consistency of generated data (quantified by the performance of the oracle model) when decreasing the noise level. Reducing $r$ also results in a degradation in the performance of the final few-shot model ($87.2\% \rightarrow 77.6\%$) corroborating that an appropriate choice of $r$ plays a crucial role in our design strategy. We investigate this further in the following paragraph.

**Effect of Noise in `GeNIe`.** We examine the impact of noise on the performance of the few-shot model in Table 4. Noise levels $r \in [0.7, 0.8]$ yield the best performance. Conversely, utilizing noise levels below 0.7 diminishes performance due to label inconsistency, as is demonstrated in Table 4 and Fig 5. As such, determining the appropriate noise level is pivotal for the performance of `GeNIe` to be able to generate challenging hard negatives while maintaining label consistency. An alternative approach to finding the optimal noise level involves using `GeNIe-Ada` to adaptively select the noise level for each source image and target class. As demonstrated in Tables 4 and A1, `GeNIe-Ada` achieves performance that is comparable to or surpasses that of `GeNIe` with fixed noise levels.

## 5  Concluding Remarks

`GeNIe`, for the first time to our knowledge, combines contradictory sources of information (a source image, and a different target category prompt) through a noise adjustment strategy into a conditional latent diffusion model to generate challenging augmentations, which can serve as hard negatives.

**Limitation.** The required time to create augmentations through `GeNIe` is on par with any typical diffusion-based competitors [4, 35]; however, this is naturally slower than traditional augmentation techniques [110, 111]. This is not a bottleneck in offline augmentation strategies, but can be considered a limiting factor in real-time scenarios. Recent studies are already mitigating this through advancements in diffusion model efficiency [87, 68, 58]. Another challenge present in any generative AI-based augmentation technique is the domain shift between the distribution of training data and the downstream context they might be used for augmentation. A possible remedy is to fine-tune the diffusion backbone on a rather small dataset from the downstream task.

**Broader Impact.** We believe ideas from `GeNIe` can have a significant impact when it comes to generating hard augmentations challenging and thus enhancing downstream tasks beyond classification. At the same time, just like any other generative model, `GeNIe` can also introduce inherent biases stemming from the training data used to build its diffusion backbone, which can reflect and amplify societal prejudices or inaccuracies. Therefore, it is crucial to carefully mitigate potential biases in generative models such as `GeNIe` to ensure a fair and ethical deployment of deep learning systems.

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

# A Appendix

## A.1 Analyzing `GeNIe`, `GeNIe-Ada`'s Class-Probabilities

The core aim of `GeNIe` and `GeNIe-Ada` is to address the failure modes of a classifier by generating *challenging* samples located near the decision boundary of each class pair, which facilitates the learning process in effectively enhancing the decision boundary between classes. As summarized in Table 4 and illustrated in Fig. 5, we have empirically corroborated that `GeNIe` and `GeNIe-Ada` can respectively produce samples $X_r, X_{r^*}$ that are negative with respect to the source image $X_S$, while semantically belonging to the class $T$. To

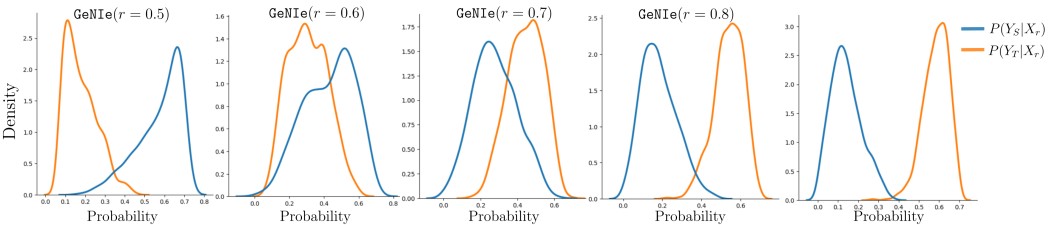

Figure A1: $P(Y_S|X_r)$ and $P(Y_T|X_r)$ for $r \in \{0.5, 0.6, 0.7, 0.8, 0.9\}$. On average, the classifier confidently predicts the source class more than the target class for $X_r$ for $r = 0.5$, and vice-versa for $r = 0.8, 0.9$. However, for $r = 0.6, 0.7$, the classifier struggles to classify $X_r$, indicating that the augmented samples are located closer to the decision boundary.

further analyze the effectiveness of `GeNIe` and `GeNIe-Ada`, we compare the source class-probabilities $P(Y_S|X_r)$ and target-class probabilities $P(Y_S|X_r)$ of augmented samples $X_r$. To compute these class probabilities, we first fit an SVM classifier (as followed in UniSiam [61]) only on the labelled support set embeddings of each episode in the *mini*Imagenet test dataset. Then, we perform inference using each episode's SVM classifier on its respective $X_r$'s and extract its class probabilities of belonging to its source class $S$ and target class $T$. These per augmentation-sample source and target class probabilities are then averaged for each episode for each $r \in \{0.5, 0.6, 0.7, 0.8, 0.9\}$ in the case of `GeNIe` and for the optimal $r = r^*$ per sample in the case of `GeNIe-Ada`, plotted as density plots in Fig. A1, Fig. A2, respectively. Fig. A1 illustrates that $P(Y_S|X_r)$ and $P(Y_T|X_r)$ have significant overlap in the case of $r \in \{0.6, 0.7\}$ indicating class-confusion for $X_r$.

Furthermore, Fig. A2 illustrates that when using the optimal $r = r^*$ found by `GeNIe-Ada` per sample, $P(Y_S|X_r)$ and $P(Y_T|X_r)$ significantly overlap around probability scores of $0.2 - 0.45$, indicating class confusion for `GeNIe-Ada` augmentations. This corroborates with our analysis in Section 4.3, Table 4 and additionally empirically proves that the augmented samples generated by `GeNIe` for $r \in \{0.6, 0.7\}$ and `GeNIe-Ada` for $r = r^*$ are actually located near the decision boundary of each class pair.

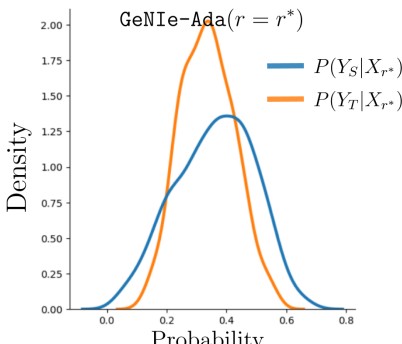

Figure A2: Significant overlap between $P(Y_S|X_{r^*})$ and $P(Y_T|X_{r^*})$ indicates high class-confusion for augmented samples generated by `GeNIe-Ada`.

## A.2 Fine-grained Few-shot Classification

To further investigate the impact of the proposed method, we compare `GeNIe` with other text-based data augmentation techniques across four distinct fine-grained datasets in a 20-way, 1-shot classification setting. We employ the pre-trained DINOV2 ViT-G [70] backbone as a feature extractor to derive features from training images. Subsequently, an SVM classifier is trained on these features, and we report the Top-1 accuracy of the model on the test set.

**Datasets:** We assess our method on several datasets: Food101 [5] with 101 classes of various foods, CUB200 [102] with 200 bird species classes, Cars196 [48] with 196 car model classes, and FGVC-Aircraft [64] with 41 aircraft manufacturer classes. We provide detailed information around fine-grained datasets in Table A2. The reported metric is the average Top-1 accuracy over 100 episodes.

Each episode involves sampling 20 classes and 1-shot from the training set, with the final model evaluated on the respective test set.

**Implementation Details:** We enhance the basic prompt by incorporating the superclass name for the fine-grained dataset: "A photo of a <target class>, a type of <superclass>". For instance, in the *food* dataset and the *burger* class, our prompt reads: "A photo of a *burger*, a type of *food*." No additional augmentation is used for generative methods in this context. We generate 19 samples for both cases of our method and also the baseline with weak augmentation.

**Results:** Table A1 summarizes the results. GeNIe helps outperform all other baselines and augmentations, including Txt2Img, by margins upto 0.5% on CUB200 [102], 6.6% on Cars196 [48], 0.1% on Food101 [5] and 5.3% on FGVC-Aircraft [64]. Notably, GeNIe exhibits great effectiveness in more challenging datasets, outperforming the baseline with traditional augmentation by about 38% for the Cars dataset and by roughly 17% for the Aircraft dataset. It can be observed here that GeNIe-Ada performs on-par with GeNIe with a fixed noise level, eliminating the necessity for noise level search in GeNIe.

Table A1: **Few-shot Learning on Fine-grained dataset:** We utilize an SVM classifier trained atop the DINOV2 ViT-G pretrained backbone, reporting Top-1 accuracy for the test set of each dataset. The baseline is an SVM trained on the same backbone using weak augmentation. Across all datasets, GeNIe surpasses this baseline.

| Method | Birds CUB200 [102] | Cars Cars196 [48] | Foods Food101 [5] | Aircraft Aircraft [64] |
|---|---|---|---|---|
| Baseline | 90.3 | 49.8 | 82.9 | 29.2 |
| Img2Img$^L$[63] | 90.7 | 50.4 | 87.4 | 31.0 |
| Img2Img$^H$[63] | 91.3 | 56.4 | 91.7 | 34.7 |
| Txt2Img[35] | 92.0 | 81.3 | 93.0 | 41.7 |
| GeNIe (r=0.5) | 92.0 | 84.6 | 91.5 | 39.8 |
| GeNIe (r=0.6) | 92.2 | 87.1 | 92.5 | 45.0 |
| GeNIe (r=0.7) | 92.5 | **87.9** | 92.9 | **47.0** |
| GeNIe (r=0.8) | 92.5 | 87.7 | **93.1** | 46.5 |
| GeNIe (r=0.9) | 92.4 | 87.1 | **93.1** | 45.7 |
| GeNIe-Ada | **92.6** | **87.9** | **93.1** | 46.9 |

Table A2: Train and test split details of the fine-grained datasets. We use the provided train set for few-shot task generation, and the provided test sets for our evaluation. For the Aircraft dataset we use manufacturer hierarchy.

| Dataset | Classes | Train samples | Test samples |
|---|---|---|---|
| CUB200 [102] | 200 | 5994 | 5794 |
| Food101 [5] | 101 | 75750 | 25250 |
| Cars [48] | 196 | 8144 | 8041 |
| Aircraft [64] | 41 | 6,667 | 3333 |

### A.3 Few-shot Classification with ResNet-34 on *tiered*Imagenet

We follow the same evaluation protocol here as mentioned in section 4.1. As summarized in Table A3, GeNIe and GeNIe-Ada outperform all other classical and generative data augmentation techniques.

### A.4 Additional details of Long-Tail experiments

We present a comprehensive version of Table 3 to benchmark the performance with different backbone architectures (e.g., ResNet50) and to compare against previous long-tail baselines; this is detailed in Table A4.

**Implementation Details of LViT:** We download the pre-trained ViT-B of LViT [107] and finetune it with Bal-BCE loss proposed therein on the augmented dataset. Training takes 2 hours on four NVIDIA RTX 3090 GPUs. We use the same hyperparameters as in [107] for finetuning: 100 epochs, $lr = 0.008$, batch size of 1024, CutMix and MixUp for the data augmentation.

**Implementation Details of VL-LTR:** We use the official code of VL-LTR [97] for our experiments. We use a pre-trained CLIP ResNet-50 backbone. We followed the hyperparameters reported in VL-

Table A3: **_tiered_-ImageNet:** Accuracies (% ± std) for 5-way, 1-shot and 5-way, 5-shot classification settings on the test-set. We compare against various SOTA supervised and unsupervised few-shot classification baselines as well as other augmentation methods, with UniSiam [61] pre-trained ResNet-34 backbone.

| ResNet-34 | | | | |
|---|---|---|---|---|
| **Augmentation** | **Method** | **Pre-training** | **1-shot** | **5-shot** |
| Weak | MAML + dist [29] | sup. | 51.7±1.8 | 70.3±1.7 |
| Weak | ProtoNet [93] | sup. | 52.0±1.2 | 72.1±1.5 |
| Weak | UniSiam + dist [61] | unsup. | 68.7±0.4 | 85.7±0.3 |
| Weak | UniSiam [61] | unsup. | 65.0±0.7 | 82.5±0.5 |
| Strong | UniSiam [61] | unsup. | 64.8±0.7 | 82.4±0.5 |
| CutMix [110] | UniSiam [61] | unsup. | 63.8±0.7 | 80.3±0.6 |
| MixUp [111] | UniSiam [61] | unsup. | 64.1±0.7 | 80.0±0.6 |
| Img2Img$^L$[63] | UniSiam [61] | unsup. | 66.1±0.7 | 83.1±0.5 |
| Img2Img$^H$[63] | UniSiam [61] | unsup. | 70.4±0.7 | 84.7±0.5 |
| Txt2Img[35] | UniSiam [61] | unsup. | 75.0±0.6 | 85.4±0.4 |
| DAFusion [100] | UniSiam [61] | unsup. | 64.1±2.1 | 82.8±1.4 |
| GeNIe (Ours) | UniSiam [61] | unsup. | **75.7±0.6** | **86.0±0.4** |
| GeNIe-Ada (Ours) | UniSiam [61] | unsup. | **76.9±0.6** | **86.3±0.2** |

LTR [97]. We augment only "Few" category and train the backbone with the VL-LTR [97] method. Training takes 4 hours on 8 NVIDIA RTX 3090 GPUs.

## A.5 More Visualizations

Additional qualitative results resembling the style presented in Fig. 4 are presented in Fig. A3, and more visuals akin to Fig. 2 can be found in Fig. A4. Moreover, we also present more visualization similar to the style in Fig. 5 in Fig. A5.

Table A4: **Long-Tailed ImageNet-LT:** We compare different augmentation methods on ImageNet-LT and report Top-1 accuracy for "Few", "Medium", and "Many" sets. † indicates results with ResNeXt50. ∗: indicates training with 384 resolution so is not directly comparable with other methods with 224 resolution. On the "Few" set and LiVT method, our augmentations improve the accuracy by 11.7 points compared to LiVT original augmentation and 4.4 points compared to Txt2Img.

| Method | Many | Med. | Few | Overall Acc |
|---|---|---|---|---|
| **ResNet-50** | | | | |
| CE [21] | 64.0 | 33.8 | 5.8 | 41.6 |
| LDAM [7] | 60.4 | 46.9 | 30.7 | 49.8 |
| c-RT [45] | 61.8 | 46.2 | 27.3 | 49.6 |
| $\tau$-Norm [45] | 59.1 | 46.9 | 30.7 | 49.4 |
| Causal [96] | 62.7 | 48.8 | 31.6 | 51.8 |
| Logit Adj. [69] | 61.1 | 47.5 | 27.6 | 50.1 |
| RIDE(4E)† [105] | 68.3 | 53.5 | 35.9 | 56.8 |
| MiSLAS [115] | 62.9 | 50.7 | 34.3 | 52.7 |
| DisAlign [112] | 61.3 | 52.2 | 31.4 | 52.9 |
| ACE† [6] | 71.7 | 54.6 | 23.5 | 56.6 |
| PaCo† [20] | 68.0 | 56.4 | 37.2 | 58.2 |
| TADE† [113] | 66.5 | **57.0** | 43.5 | 58.8 |
| TSC [56] | 63.5 | 49.7 | 30.4 | 52.4 |
| GCL [55] | 63.0 | 52.7 | 37.1 | 54.5 |
| TLC [50] | 68.9 | 55.7 | 40.8 | 55.1 |
| BCL† [117] | 67.6 | 54.6 | 36.6 | 57.2 |
| NCL [52] | 67.3 | 55.4 | 39.0 | 57.7 |
| SAFA [38] | 63.8 | 49.9 | 33.4 | 53.1 |
| DOC [104] | 65.1 | 52.8 | 34.2 | 55.0 |
| DLSA [106] | 67.8 | 54.5 | 38.8 | 57.5 |
| ResLT [18] | 63.3 | 53.3 | 40.3 | 55.1 |
| PaCo [19] | 68.2 | 58.7 | 41.0 | 60.0 |
| LWS [44] | 62.2 | 48.6 | 31.8 | 51.5 |
| Zero-shot CLIP [76] | 60.8 | 59.3 | 58.6 | 59.8 |
| DRO-LT [85] | 64.0 | 49.8 | 33.1 | 53.5 |
| VL-LTR [97] | 77.8 | 67.0 | 50.8 | 70.1 |
| Cap2Aug [83] | 78.5 | **67.7** | 51.9 | 70.9 |
| GeNIe-Ada | **79.2** | 64.6 | **59.5** | **71.5** |
| **ViT-B** | | | | |
| LiVT* [107] | 76.4 | 59.7 | 42.7 | 63.8 |
| ViT [24] | 50.5 | 23.5 | 6.9 | 31.6 |
| MAE [33] | 74.7 | 48.2 | 19.4 | 54.5 |
| DeiT [99] | 70.4 | 40.9 | 12.8 | 48.4 |
| LiVT [107] | 73.6 | 56.4 | 41.0 | 60.9 |
| LiVT + Img2Img$^L$ | 74.3 | 56.4 | 34.3 | 60.5 |
| LiVT + Img2Img$^H$ | 73.8 | 56.4 | 45.3 | 61.6 |
| LiVT + Txt2Img | **74.9** | 55.6 | 48.3 | 62.2 |
| LiVT + GeNIe (r=0.8) | 74.5 | 56.7 | 50.9 | 62.8 |
| LiVT + GeNIe-Ada | 74.0 | **56.9** | **52.7** | **63.1** |

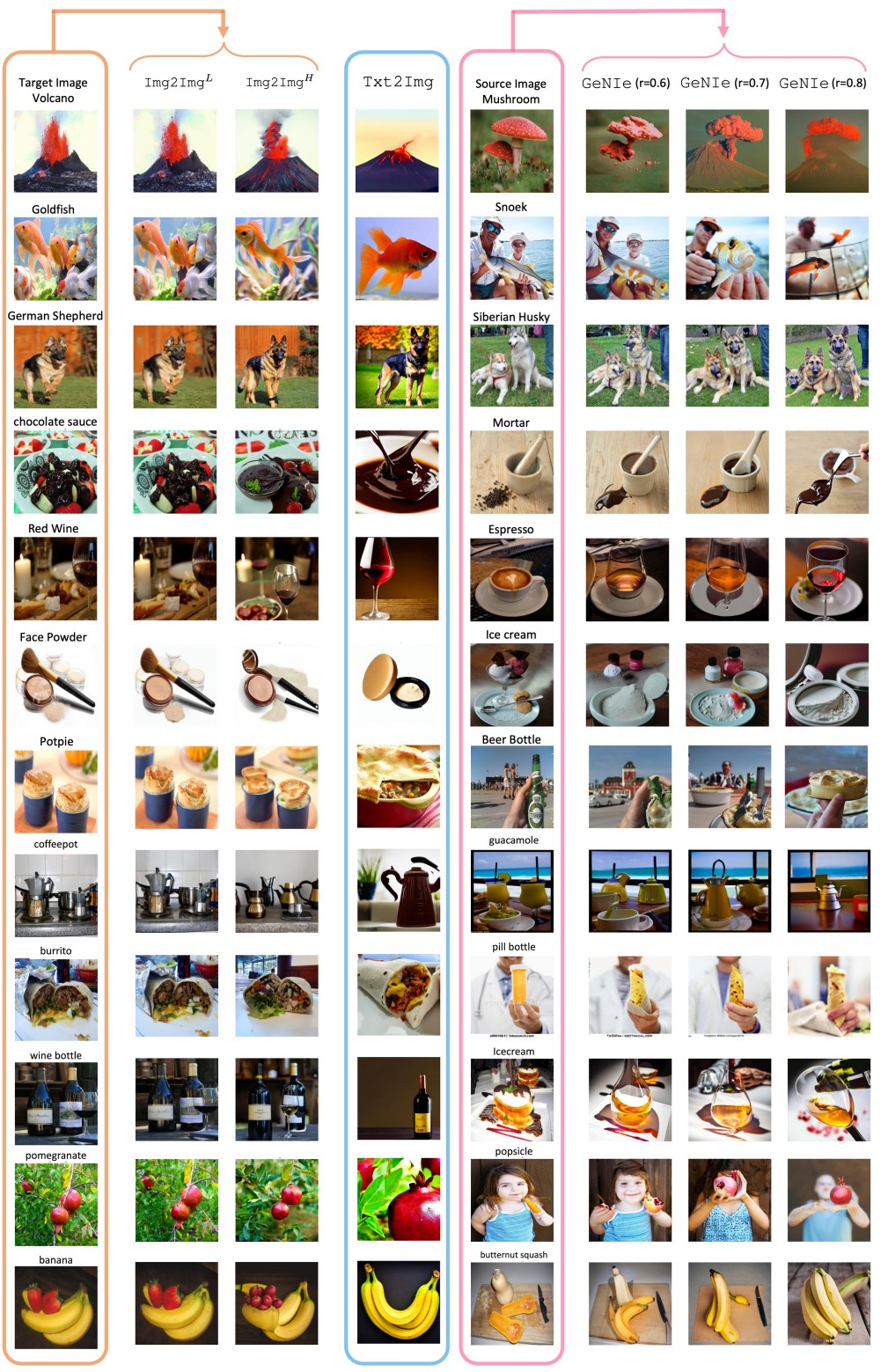

Figure A3: **Visualization of Generative Samples:** More visualization akin to Fig. 4. We compare `GeNIe` with two baselines: `Img2Img`$^L$ **augmentation** uses both image and text prompt from the same category, resulting in less challenging examples. `Txt2Img` **augmentation** generates images based solely on a text prompt, potentially deviating from the task's visual domain. `GeNIe` **augmentation** incorporates the target category name in the text prompt along with the source image, producing desired images with an optimal amount of noise, and balancing the impact of the source image and text prompt.

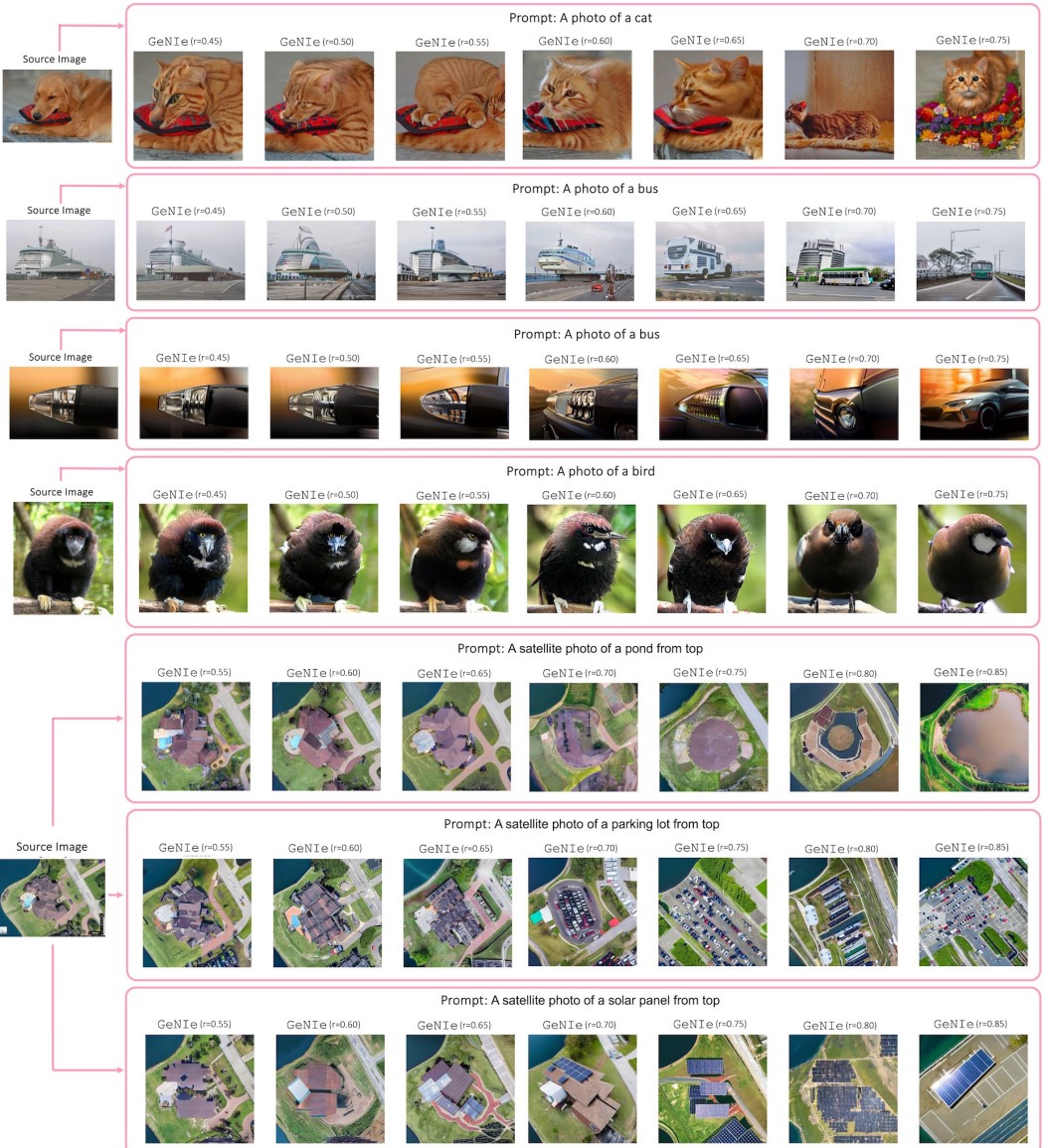

Figure A4: **Effect of noise in GeNIe:** Akin to Fig. 2, we use GeNIe to create augmentations with varying noise levels. As is illustrated in the examples above, a reduced amount of noise leads to images closely mirroring the semantics of the source images, causing a misalignment with the intended target label.

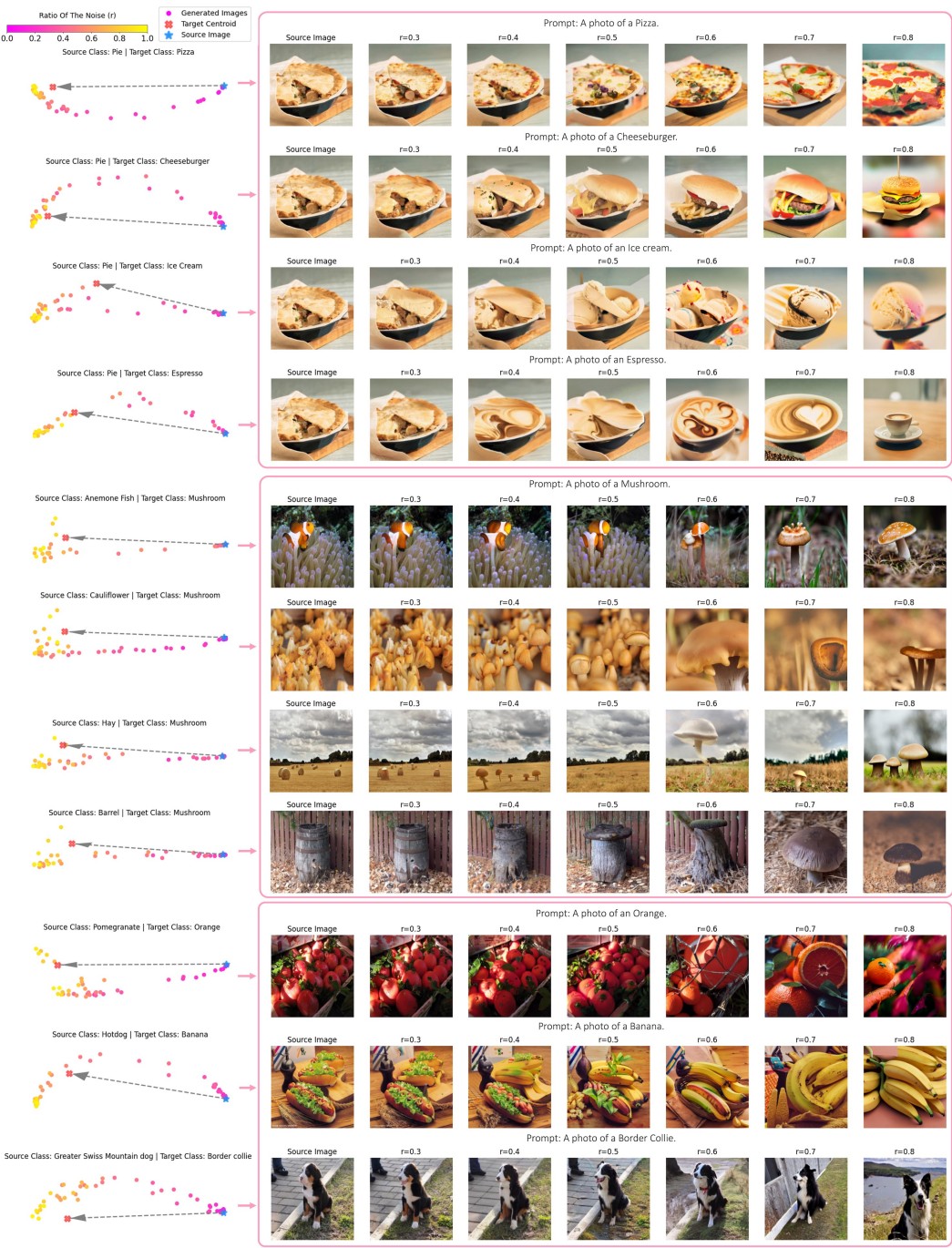

Figure A5: **Effect of noise in** `GeNIe`: Similar to Fig. 5, we pass all the generated augmentations through the DinoV2 ViT-G model, which acts as our oracle model, to obtain their associated embeddings. Subsequently, we employ PCA for visualization purposes. The visualization reveals that the magnitude of semantic transformations is contingent upon both the source image and the specified target category.

