# OpenReview forum: "GeNIe: Generative Hard Negative Images Through Diffusion"
_NeurIPS.cc/2024/Conference — Submitted to NeurIPS 2024_

### Official Review · Reviewer_hT8y · 2024-07-06

**Soundness:** 4
**Presentation:** 4
**Contribution:** 2
**Rating:** 6
**Confidence:** 4

**Summary:**

The paper proposes to employ text-to-image latent diffusion models to augment images through a controlled modification such that the resultant class is different from the source class. Such augmented images are referred to as hard negative images. Building upon SDEdit style image modification, the paper controls the extent of modification by adaptively determining the appropriate noise-scale for each image separately. The benefits of this type of augmentation have been demonstrated on few-shot and long-tailed imagenet classification tasks.

**Strengths:**

- The paper is very well written presenting the core idea of generating hard-negative images by modifying an image with a caption of another class. This idea is simple, intuitive and interesting.
- Furthermore, the algorithm to determine the optimal noise-level for each image adaptively is not only simple and intuitive but also effective in eliminating dependence on hyperparameters. I feel that a connection can be made to the recent work [1] on phase-transition in diffusion models since this algorithm is attempting to find the diffusion-time when phase-transition occurs.
- The evaluation is comprehensive considering a variety of diffusion-augmentation baselines as well as traditional augmentations.
- The paper illustrates the effectiveness of the adaptive search procedure through separate experiments with DINO-v2 and visualisations.
- In many cases, synthetic data generation with a diffusion model may be replaced by a simpler retrieval baseline [2]. However, the goal of this work is to use a diffusion model to search for and generate hard negatives, which is an interesting deviation from some of the previous synthetic data augmentation approaches.

[1] Sclocchi, Antonio, Alessandro Favero, and Matthieu Wyart. "A phase transition in diffusion models reveals the hierarchical nature of data." arXiv preprint arXiv:2402.16991 (2024).

**Weaknesses:**

- From the various results in the paper, it seems that the Text2Image, GeNIe, and GeNIe-Ada achieve comparable performance with respect to each other on average. This seems to suggest that the majority of the gains can be attributed to the increased number of _distinct_ examples --- as compared to regular augmentations which simply apply different transformations to the same image --- for each class rather than the hard-negatives in GeNIe/GeNIe-Ada.
- Additionally, it seems that beyond some threshold, any value of $r$ that changes the source-image to the target image yields comparable performance indicating that it may be sufficient to generate an augmentation that is similar to source-image and it need not specifically be a _hard-negative_.  It may be useful to consider some other applications where images lying in the boundary of the classifier may be informative: for example, see recent work on generating outliers [1] for OOD detection.
- GeNIe-Ada algorithm is compute-intensive as compared to a simple Text2Image augmentation since it requires generating several augmentations for each source image before selecting one optimal augmentation that lies on the decision boundary. Given how close the text2image and genie-ada performances are in some cases, it may be possible that we could generate more augmentations using text2image in the same compute budget and improve over GeNIe.
- (minor) GeNIe is applicable to the fine-tuning stage rather than the pretraining stage.

[1] Du, X., Sun, Y., Zhu, J. and Li, Y. Dream the impossible: Outlier imagination with diffusion models. NeurIPS 2024.

**Questions:**

See weaknesses.
- How can we apply GeNIe for classification involving a larger number of classes? Could you please elaborate on this statement in the paper: _For GeNIe, instead of randomly sampling the source images from other classes, we use a confusion matrix on the training data to find the top-4 most confused classes and only consider those classes for random sampling of the source image. The source category may be from “Many”, “Med”, or “Few sets”._ If the training converges, how can the confusion matrix on train data be informative?
- If the class of the source image and target class are not semantically compatible, how does the modification look like?
- Do you have any examples of failure cases of algorithm 1?

**Limitations:**

Yes, limitations are addressed.

---

> ### Author Rebuttal · Authors · 2024-08-06
>
> **[hT8y][W1]: seemingly comparable performance of Txt2Img and GeNIe:**
>
> - Thanks for this comment. Indeed diffusion-based augmentation techniques offer a notable margin compared to traditional approaches (such as Cutmix and Mixup). We offer an enhancement of a diffusion based approach by proposing GeNIe and further automating and enhancing it through GeNIe-Ada.
>
> - Besides, please note that GeNIe and GeNIe-Ada almost consistently outperform txt2Img, a few examples summarized in the following: (i) Few-shot classification scenarios, as demonstrated in Section 4.1 - Table 1 and Table 2. On miniImagenet, GeNIe and GeNIe-Ada offer improvements in the range of 1.3% - 3.1%, and in the range of 0.8% - 0.9% on tieredImagenet. (ii) Fine-grained classification, as demonstrated in Section A2 - Table A1. Here, GeNIe and GeNIe-Ada offer improvements in the range of 5.3% on the Aircrafts dataset and 6.3% on Cars196.
>
> - Furthermore, our newest experiments in Table X1 and X2 (attached PDF) further corroborate the superiority of GeNIe compared to Txt2Img when adopting a more recent and smaller diffusion models (i.e. Stable Diffusion v3, SDXL-Turbo).
>
>
> **[hT8y][W2]: consistently choosing a large $r$. Considering other application such as OOD.:**
>
>
> - The Fig. 6 and Table 4 of Section 4.2 combined tell the story a bit better. Even though it might seem a larger $r$ would be a sensible option to opt for (at least on average), in practice such a choice will distance the model from capturing the context and low-level features of the source category. So, striking a balance between as low as possible choice of $r$ (to capture low-level source features) and as high as possible to result in a high Oracle score (confirming the correct target category) is crucial, and this is what GeNIe-Ada does.
>
> - Note that the margin between GeNIe-Ada and GeNIe (for instance, with $r\geq 0.9$) becomes more pronounced when taking the $1$-shot setting into account, in all few-shot settings.
>
> - Considering GeNIe for OOD applications is a great idea, and we have already started looking into it, following your remark. While we will definitely cite [Your Ref 1, on OOD] in the revised paper, we feel this requires more work than we can afford for this revision and would fit best into our future work.
>
>
> **[hT8y][W3]: Txt2Img with lager number of augmentations to make up for its relatively lower computation complexity:**
>
> - Thanks for the insightful remark. Given that GeNIe-Ada searches for the best hard-negative between multiple noise-ratios $r$'s, it naturally requires a higher compute budget than Txt2Img that only uses $r=1$. For this experiment, we use GeNIe-Ada with $r \in \{0.6, 0.7, 0.8\}$ to compare with Txt2Img. Based on this, we only have $3$ paths, with steps of $0.1$), and for each of which we go through partial reverse diffusion process. E.g. for $r=0.6$ we do $30$ steps instead of standard $50$ steps of Stable Diffusion. This practically breaks down the total run-time of GeNIe-Ada to approximately $2$ times that of the standard reverse diffusion (GeNIe-Ada: total $r = 0.6 + 0.7 + 0.8 = 2.1$ vs Txt2Img total $r = 1$). Thus, to be fair, we generate twice as many Txt2Img augmentations as compared to GeNIe-Ada to keep a constant compute budget across the methods, following your suggestion. The results are shown in Table X3. As can be seen, even in this new setting, GeNIe-Ada offers a performance improvement of $0.8\%$ to $1.9\%$ across different backbones.
>
> - Note that GeNIe itself (due to partial reverse process) is actually faster than txt2Img, and yet consistently more accurate.
>
> **[hT8y][W4]: GeNIe is applicable to the fine-tuning stage rather than the pretraining stage:**
>
> - We agree, even though theoretically nothing stops one from applying GeNIe on a pretraining stage, especially in an offline augmentation setting where latency is of no concern. Note that, data augmentation is mostly beneficial in data deficient settings than the pretraining stage where data is typically abundant.
>
> **[hT8y][Q1-p1]: classification involving a larger number of classes:**
>
> - GeNIe can be applied to large number of classes, as we already do this in the long-tail ImageNet experiment (Section 4.2). However, as discussed in the paper, we use the confusion matrix to sub-select that source-target pairs in an efficient manner out of the large pool of classes.
>
> **[hT8y][Q1-p2]: using confusion matrix post training convergence:**
>
> - Thanks for highlighting this. In our ImageNet-LT experiments, we compute the confusion matrix on a held-out set split from the full training set. We will clarify this in the revised draft.
>
>
> **[hT8y][Q2]: incompatible source-target classes:**
>
> - Thanks for this comment. Semantic compatibility is indeed a difficult matter to quantify. That said, following your remark, Fig. X1 (see attached PDF) summarizes a few examples where source and target are incompatible. As can be seen, GeNIe starts from pizza and gradually transforms the image towards the volcano while capturing the low-level features of the pizza image. This transformation (semantic switch), however, in such incompatible cases seems to occur at larger noise ratios $r$ when compared to easier cases. We will include this figure in the camera ready version and elaborate on this point in the main text of the revised draft.
>
> **[hT8y][Q3]: failure cases of the algorithm:**
>
> - Great point. We are conscious that there are _exceptional_ failure cases (where the right choice of $r$ is ambiguous due to the presence of a mixture of both source and target predominant visual features - e.g in Fig. X1, the top row) in which the automated noise selection process does not return an ideal outcome. We will discuss this point further in the Limitations Section in the revised draft.
>
> We do hope this addresses all your concerns, please do not hesitate to let us know if you have any further suggestions.

---

> > ### Comment · Reviewer_hT8y · 2024-08-09
> > **Thank you!**
> >
> > Thank you very much for your additional experiments and responses; I have raised my rating supporting acceptance!

---

> > > ### Author Response · Authors · 2024-08-10
> > > **Much Appreciated!**
> > >
> > > We are pleased that you are happy with our additional experiments, and responses. Thank you for raising your score!

---

### Official Review · Reviewer_3KAK · 2024-07-11

**Soundness:** 2
**Presentation:** 3
**Contribution:** 3
**Rating:** 6
**Confidence:** 4

**Summary:**

This paper introduces GeNIe, a data augmentation method for training vision models using synthetic images. GeNIe generates images by combining a source category image with a target category text prompt, selecting those that feature source characteristics but belong to the target category as negative samples. Experimental results show that GeNIe improves performance in both few-shot and long-tail distribution settings.

**Strengths:**

* The proposed GeNIe improves the performance in few-shot and long-tail distribution settings.
* The paper provides extensive experiments to support the claims,  including the selection of noise levels.
* The paper is well-written and easy to follow.

**Weaknesses:**

The key idea of GeNIe is to use image editing to combine features from two categories. Here are several questions:

* Regarding controllable image augmentation
  * Line 9 mentions that GeNIe "retains low-level and background features from the source image." How does GeNIe control which features are retained or changed?
  * To combine features from different categories, how about adding the attribute from the target category to the prompt? For example, a "[dog] with [wings]".  This method does not require carefully selection of denoise steps.
  * Other image editing methods, such as those in [1] and [2], efficiently control image changes using prompts or user instructions.  For example, they can transform a car into a motorcycle in Figure 2, while keeping the background unchanged for more challenging negative samples. What advantages does GeNIe offer over these methods?



* GeNIe generates images "using images from all other classes as the source image" (line 227). Will all (source image, target prompt) pairs lead to effective image generation? Which types of pairs contribute the most to the final accuracy?

     [1] Prompt-to-Prompt Image Editing with Cross-Attention Control

     [2] InstructPix2Pix: Learning to Follow Image Editing Instructions, CVPR 2023

**Questions:**

Please refer to the above weaknesses.

**Limitations:**

The paper has discussed the limitations.

---

> ### Author Rebuttal · Authors · 2024-08-06
>
> $\textbf{[3KAK][W1]}$: $\textbf{How does $\texttt{GeNIe}$ control which features are retained or changed}$:
>
>
> We instruct the diffusion model to generate an image by combining the latent noise of the source image with the textual prompt of the target category. This combination is controlled by the amount of added noise and the number of reverse diffusion iterations. This approach aims to produce an image that aligns closely with the semantics of the target category while preserving the background and features from the source image that are unrelated to the target.
>
> - To demonstrate this, we have prepapred Fig X3 in the attached PDF. Here, we are progessivley moving towards the two key components of $\texttt{GeNIe}$: (i) careful choice of $r$ and (ii) contradictory prompt. The input image is a bird in a cage. The top row shows a Stable Diffusion model, unprompted. As can be seen, such a model can generate anything (irrespective of the input image) with a large $r$. Now prompting the same model with "a photo of a bird" allows the model to preserve low-level and contextual features of the input image (up to $r = 0.8$ and $0.9$), then for $r = 1.0$ it returns a bird but the context has nothing to do with the source input. This illustrates how a careful choice of $r$ can help preserve such low-level features, and is a key idea behind $\texttt{GeNIe}$. However, we also need a semantic switch to a different target class as shown in the last row where a hardly seen image of a dog in a cage is generated by a combination of a careful choice of $r$ and the contradictory prompt, leading to the full mechanics of $\texttt{GeNIe}$. This sample now serves as hard negative for the source image (bird class).
>
>
> $\textbf{[3KAK][W2]}$: $\textbf{adding attributes from the target category ... a ``[dog] with [wings]''}$:
>
> - Based on the reviewer's example, it appears they are suggesting adding attributes from the source category to the prompt. We should clarify the distinction between ambiguous examples and hard examples. For instance, using a prompt like "[dog] with [wings]" results in an image of a dog with wings (please see Fig. X2-[C]), but the label for this example remains unclear due to its ambiguity. Such ambiguous examples could potentially confuse the training process. In contrast, hard examples are those where the label is clearly defined. For example, a prompt like "dog in a cage" provides a clear context and should be labeled as "dog" for an animal classification task. We aim to generate these hard examples, where the correct label is unambiguous, to improve the clarity and effectiveness of the training.
>
> - We suspect the main point from the reviewer's remark is to asses whether the design engineering of $\texttt{GeNIe}$ can be replaced by a standard Stable Diffusion model given a more elaborate prompt reflecting on the low-level contextual information of the source image? This is a great suggestion, even though we need to highlight that giving a contradictory prompt opposing the source image is by itself part of $\texttt{GeNIe}$'s proposition and novelty. That said, Fig. X2 does exactly that: comparing a standard Stable Diffusion with a more elaborate prompt "a dog in a cage" and $\texttt{GeNIe}$. As can be seen, the former can result in dog in a cage where neither dog nor the cage resemble those in the source image. On the contrary, $\texttt{GeNIe}$ does preserve the contextual features of the source image, as such generating effective/challenging hard negatives for the given source image.
>
> $\textbf{[3KAK][W3]}$: $\textbf{advantage of $\texttt{GeNIe}$ over image editing approaches in hard negative generation}$:
>
> - According to our main contributions stated in line 57 of the main manuscript, we do not claim novelty on using diffusion models for image editing. Instead, our primary contribution lies in leveraging these tools to generate $\textit{hard negatives}$ for training. Note that advances in image editing techniques ([Your Ref 1] and [Your Ref 2]) is orthogonal to our contribution, and we believe that improvement in image editing techniques over time, could further enhance our results and increase the effectiveness of our approach. In that light, other diffusion based image editing techniques can also be used as the backbone engine of $\texttt{GeNIe}$, where the novelty is (i) to find the right noise threshold and (ii) to provide a contradictory prompt.
> - Notably, we will cite both [Your Ref 1] and [Your Ref 2] in the revised draft as alternatives for an image editing backbone.
>
>
>
>
> $\textbf{[3KAK][W4]}$: $\textbf{choice/effectiveness of source-target class pairs}$:
>
> - You are right. Not all pairs will lead to examples informative for training. For the very same reason, in Lines 136 and 258, we discuss choosing the pairs of source and target using the confusion matrix of an initially trained model (e.g. for long-tail distributions).
>
> We do hope this addresses all your concerns, please do not hesitate to let us know if you have any further suggestions.

---

> > ### Comment · Reviewer_3KAK · 2024-08-12
> >
> > Thanks to the authors for the rebuttal, which addressed my concerns well. I will be increasing my rating.

---

> > > ### Author Response · Authors · 2024-08-12
> > > **Much appreciated**
> > >
> > > Dear Reviewer 3KAK,
> > >
> > > We are pleased to hear we've have successfully addressed your concerns; many thanks for raising your final score.
> > >
> > > Best regards,
> > > Authors

---

### Official Review · Reviewer_Q6Ew · 2024-07-12

**Soundness:** 3
**Presentation:** 3
**Contribution:** 3
**Rating:** 6
**Confidence:** 3

**Summary:**

In this paper, the idea is to generate data for data augmentation by utilizing a pre-trained diffusion model. The method employs different text prompts and an adjusted noise scheduler to generate hard negative samples for the source distribution. "GeNIe" creates new augmentations using diffusion by leveraging source images and contradictory target prompts. "GeNIe-Ada" adjusts noise levels on a per-sample basis, using the classifier as the condition boundary to select the right threshold.

**Strengths:**

- The method offers infinite possibilities to separate the source from the target.
- The idea is simple, original, and convincing.
- The ablation studies and experiments demonstrate strong performance.

**Weaknesses:**

- The method is slow, particularly GeNIe-Ada, as it requires generating an image through multiple forward passes of a diffusion model and using a classifier to select the appropriate threshold $r$.

- The number of steps required to retain low-level features is crucial for optimizing the method's performance.

- The method relies on access to a foundational text-to-image model trained on billions of images.

**Questions:**

- Stable diffusion utilizes data scraped from the web (LAION), and there is a high probability that the image from validation set of ImageNet is included in the LAION training set. Moreover, stable diffusion tends to replicate the training set [1]. How do you ensure that the augmented images used are not also present in our testing set?

- How does the method perform when using a diffusion model trained exclusively on ImageNet or other diffusion models besides Stable Diffusion?

- How do you ensure that the generated images still resemble natural images? Some prompts could diverge significantly from the source distribution, resulting in images that may be far from the original distribution.

[1] Nicholas Carlini et al.: "Extracting Training Data from Diffusion Models"

**Limitations:**

/

---

> ### Author Rebuttal · Authors · 2024-08-06
>
> $\textbf{[Q6Ew][W1]}$: $\textbf{slowness of \texttt{GeNIe-Ada}}$:
>
> - Thanks for this remark. As we highlight in our limitations, we acknowledge that $\texttt{GeNIe}$ is comparatively slower than traditional augmentation methods, while standing on par with (or even faster in the case of barebone $\texttt{GeNIe}$, due to partial reverse process) other generative methods. Optimization/Efficiency of diffusion based models is active line of research making approaches like our more favorable in the future due to their superior in performance and capacity.
>
> - When it comes to $\texttt{GeNIe-Ada}$, in practice we only probe $r \in [0.5, 0.9]$ (means $5$ paths, with steps of $0.1$), and for each of which we go through partial reverse diffusion process. E.g. for $r=0.5$ we do $25$ steps instead of standard $50$ steps of Stable Diffusion. This practically breaks down the total run-time of $\texttt{GeNIe-Ada}$ to roughly $2-3$ times any standard reverse diffusion.
>
> - Lastly, as discussed in Section 5, this latency might be even irrelevant/negligible when it comes to offline augmentation scenarios.
>
>
> $\textbf{[Q6Ew][W2]}$: $\textbf{crucial importance of $r$}$:
>
> - We agree, and that is the reason behind automating this critical parameter (through $\texttt{GeNIe-Ada}$) in favor of efficiency.
>
>
> $\textbf{[Q6Ew][W3]}$: $\textbf{relying on accessing a foundation model}$:
>
> - We believe with the current upsurge of interest, such foundation models are going to become commodities in the near future, further capitalizing on the importance of methodologies like ours when it comes to data augmentation. Notably, $\texttt{GeNIe}$ outperforms other Diffusion based competitors adopting similar engine as we illustrate in Sections 4.1 and 4.2.
>
> $\textbf{[Q6Ew][Q1]}$: $\textbf{ensuring augmented samples are not present in the test set}$:
>
> - This a great comment. To substantiate our understanding, we have a run set of experiments on the few-shot setting. To set the scene, we use pretrained image encoder as an oracle (DeiT-Base) to extract the latent embeddings corresponding to the train (i.e. support) set, test (i.e. query set) and augmentations generated by $\texttt{GeNIe}$. Fig. X4 demonstrates the distribution of distances between train-test and augmentation-test pairs across $600$ episodes. As can be seen, the (mean of the) distribution of augmentation-test pair is higher that that of train-test pair indicating that the augmented samples are indeed different from the test sets (based on the strong assumption of train and test sets being mutually exclusive). This is further illustrated in the last column of Fig. X4 on a UMAP embedding plot of a random episode where the embedding of train, test and augmentations are plotted. Here again there is noticeable separation between the augmentation and test samples as compared to train and test samples.
>
> $\textbf{[Q6Ew][Q2]}$: $\textbf{performance on a different diffusion model or one trained on ImageNet}$:
>
> - Using much larger dataset such as LAION seems to be the prevalent choice for training Diffusion models. We had two concerns in not adopting models based on ImageNet: (i) to avoid limiting the taxonomy of classes to the ones present in Imagenet, but a much larger set (present in LAION); (ii) also considering the fact that some of our few-shot settings (e.g. tiered/mini-ImageNet) are derivatives of ImageNet itself.
>
> - Following your suggestion, we have tried experimenting with both smaller as well as more recent diffusion models (please see the PDF, therein Table X1 and X2). More specifically, we have used Stable Diffusion XL-Turbo to generate hard-negatives through $\texttt{GeNIe}$ and $\texttt{GeNIe-Ada}$. Few-shot classification results on miniImagenet with these augmentations are shown in Table X1. The accuracies follow a similar trend to that of Table 1 in the main manuscript, where Stable Diffusion 1.5 was used to generate augmentations. $\texttt{GeNIe-Ada}$ improves UniSiam's few-shot performance the most as compared to $\texttt{GeNIe}$ with different noise ratios $r$, and even when compared to $\texttt{Txt2Img}$. This empirically indicates the robustness of $\texttt{GeNIe}$ and $\texttt{GeNIe-Ada}$ to different diffusion engines. Note that, Stable Diffusion XL-Turbo by default uses $4$ steps for the sake of optimization, and to ensure we can have the right granularity for the choice of $r$ we have set the number of steps to $10$. That is already 5 times faster than the standard Stable Diffusion v1.5 with $50$ steps used through the original submission. Our experiments with Stable Diffusion v3 (which is a totally different model with a Transformers backbone) also in Table X2 also convey the same message. As such, we believe our approach is generalizable across different diffusion models.  It is to be noted that for SDv3.0 (Table X2), we test the few-shot classification accuracies on 200 episodes instead of the standard setting following 600 episodes. This leads to a higher standard deviation in the reported accuracy scores. We will report accuracy scores using the complete 600 episodes in the final draft of the camera ready version.
>
>
> $\textbf{[Q6Ew][Q3]}$: $\textbf{ensuring generated images resemble natural ones}$:
>
> - We agree that it is almost impossible to ensure every generated image necessarily lies in the manifold of natural images; however, the generated imaged are definitely closer to natural images when compared to traditional methods such as Cutmix and Mixup. We will modify the text to reflect on this accordingly.
>
> - Regardless of generated augmentation lying in the manifold of natural images, we demonstrate the efficacy of $\texttt{GeNIe}$ on $7$ different datasets, irrespective of whether or not the downstream datasets have semantic overlap with the pretraining data of the diffusion model.
>
> We do hope this addresses all your concerns, please do not hesitate to let us know if you have any further suggestions.

---

> > ### Author Response · Authors · 2024-08-12
> > **Follow Up - Deadline Approaching**
> >
> > Dear Reviewer Q6Ew,
> >
> > Firstly, many thanks for your rigorous, positive and constructive feedback. The deadline for reviewer-author discussion is approaching soon. We have put in tremendous effort in compiling a detailed response trying our very best to address all your concerns (please see the attached PDF and our P2P responses). If you are convinced and happy with our responses, please kindly consider re-evaluating/raising your *final score*; please also let us know if you have any further questions or concerns; we'll be more than happy to address those.
> >
> > Many thanks for your insightful feedback.
> >
> > Best regards, Authors.

---

### Official Review · Reviewer_qsxy · 2024-07-13

**Soundness:** 4
**Presentation:** 4
**Contribution:** 3
**Rating:** 7
**Confidence:** 3

**Summary:**

This paper introduces a novel augmentation method based on diffusion models. A latent diffusion model conditioned on a text prompt generates hard negatives, by adjusting the noise level. The hard negatives can be used as challenging augmentations. The authors demonstrate the effectiveness of their approach on long-tail and few-shot settings.

**Strengths:**

- Well-written paper with clear contributions and presentation.
- Extensive experiments and evaluation.
- Interesting and useful idea.
- Code included in the supplementary.

**Weaknesses:**

I am generally happy with the paper, experiments, and presentation. A weakness seems to be the selection of the noise ratio r.  The authors propose an algorithm for this. However, I am concerned how sensitive it is for different datasets or classification settings. This might affect performance in other settings or in real-world scenarios. If this is true, it might degrade the overall method's usefulness.

**Questions:**

Can the authors comment on the above weakness, regarding the selection of r and its sensitivity to the dataset or setting?

Also, have the authors considered a different latent diffusion model? Would a smaller diffusion model and/or trained on a smaller amount of data give similar results/benefit?

**Limitations:**

The authors have added a section for limitations and a section for broader impact.

---

> ### Author Rebuttal · Authors · 2024-08-06
>
> $\textbf{[qsxy][W1]}$: "$\textit{happy with the paper, experiments, and presentation}$", $\textbf{selection of noise ratio $r$ across different datasets}$:
>
> - We are pleased with reviewer's positive feedback, also for finding our proposed ideas interesting.
>
> - Thanks for the interesting remark. Indeed, in Appendix A2 Table A1 we provide further experimentation on fine-grained classification (as an example of other $\textit{benchmarks}$) which we think corroborates that $\texttt{GeNIe-Ada}$ can handle such unforeseen circumstances to a good extent. Notably, throughout the paper we investigate the impact of GeNIe-Ada in $7$ different $\textit{datasets}$(tiered and miniImagenet, CUB200, Cars196, Imagenet-LT, Food101, Aircraft) demonstrating the robustness of $\texttt{GeNIe-Ada}$ across datasets. That said, we are conscious that there are $\textit{exceptional}$ failure cases (where the right choice of $r$ is ambiguous due to the presence of a mixture of both source and target predominant visual features - Fig. X1, top row) in which the automated noise selection process does not return an ideal outcome, as is already explained in response to reviewer $\textbf{hT8y}$. We will discuss this point further in our revised draft.
>
>
> $\textbf{[qsxy][Q1]}$: $\textbf{different diffusion model and/or smaller dataset}$:
>
>
> - Regarding smaller dataset, the smallest dataset we are aware of adopted for training diffusion models is ImageNet, and using much larger dataset such as LAION seems to be the prevalent choice. We had two concerns in not adopting ImageNet: (i) to avoid limiting the taxonomy of classes to the ones present in Imagenet, but a much larger set (present in LAION); (ii) also considering the fact that some of our few-shot settings (e.g. tiered/mini-ImageNet) are derivatives of ImageNet itself.
>
> - Following suggestion, we have tried experimenting with both smaller as well as more recent diffusion models (please see the PDF, therein Table X1 and X2). More specifically, we have used Stable Diffusion XL-Turbo to generate hard-negatives through $\texttt{GeNIe}$ and $\texttt{GeNIe-Ada}$. Few-shot classification results on miniImagenet with these augmentations are shown in Table X1. The accuracies follow a similar trend to that of Table 1 in the main manuscript, where Stable Diffusion 1.5 was used to generate augmentations. $\texttt{GeNIe-Ada}$ improves UniSiam's few-shot performance the most as compared to $\texttt{GeNIe}$ with different noise ratios $r$, and even when compared to $\texttt{Txt2Img}$. This empirically indicates the robustness of $\texttt{GeNIe}$ and $\texttt{GeNIe-Ada}$ to different diffusion engines. Note that, Stable Diffusion XL-Turbo by default uses $4$ steps for the sake of optimization, and to ensure we can have the right granularity for the choice of $r$ we have set the number of steps to $10$. That is already 5 times faster than the standard Stable Diffusion v1.5 with $50$ steps used through the original submission. Our experiments with Stable Diffusion v3 (which is a totally different model with a Transformers backbone) also in Table X2 convey the same message. As such, we believe our approach is generalizable across different diffusion models.
>
> It is to be noted that for SDv3.0 (Table X2), we test the few-shot classification accuracies on 200 episodes instead of the standard setting following 600 episodes. This leads to a higher standard deviation in the reported accuracy scores. We will report accuracy scores using the complete 600 episodes in the final draft of the camera ready version.
>
> We do hope this addresses all your concerns, please do not hesitate to let us know if you have any further suggestions.

---

> > ### Author Response · Authors · 2024-08-12
> > **Follow-up on our responses**
> >
> > Dear Reviewer qsxy,
> >
> > Firstly, many thanks for your rigorous, positive and constructive feedback. The deadline for reviewer-author discussion is approaching soon. We have put in tremendous effort in compiling a detailed response trying our very best to address all your concerns (please see the attached PDF and our P2P responses). If you are convinced and happy with our responses, please kindly consider re-evaluating/raising your *final score*; please also let us know if you have any further questions or concerns; we'll be more than happy to address those.
> >
> > Once again, thank you for your insightful feedback.
> >
> > Best regards, Authors.

---

### Author Rebuttal · Authors · 2024-08-06

We do appreciate reviewer's constructive feedback which helped to further improve the quality and clarity of the paper. We are please by the positive feedback from reviewers [$\textbf{qsxy}$ and $\textbf{Q6Ew}$] for finding our proposed ideas "interesting", "original" and "convincing"; we also thank $\textbf{ALL reviewers}$ for finding our narrative "well-written" and our experimentation and ablation studies "extensive/comprehensive" and supportive of the proposed ideas.

After perusing reviewer's remarks and recommendations, we have put in tremendous effort to provide further evidence (new experimentation and qualitative demonstrations) to corroborate the efficacy of $\texttt{GeNIe}$ as summarized below:

- In response to reviewer's [$\textbf{qsxy}$ and $\textbf{Q6Ew}$] we present two sets of new experimentation with $\textit{different}$ and $\textit{smaller}$ Diffusion models.

- In response to reviewer $\textbf{Q6Ew}$, regarding ensuring discrepancy between the generated augmentations and the test set, we present a statistical results on image embeddings.

- In response to reviewer $\textbf{3KAK}$, we present two new sets of qualitative results elaborating on how $\texttt{GeNIe}$ preserves low-level features as well as on the impact of using more elaborate prompt instead of $\texttt{GeNIe}$.

- In response to reviewer $\textbf{hT8y}$, we present new sets of results on comparing $\texttt{txt2Img}$ with larger number of augmentations and $\texttt{GeNIe}$, and qualitative results on source-target incompatibility as well as potential failure cases of $\texttt{GeNIe-Ada}$.

$\textbf{Remark}$: Please find attached a PDF summarizing all our new experimentation and qualitative demonstrations. We will be referring to this PDF throughout our point-to-pint response to each reviewer.

We do hope this addresses reviewer's concerns and questions, and look forward to engaging further during reviewer-author discussion period.

---

### Decision · Program_Chairs · 2024-09-25

**Decision:**

Reject

**Comment:**

This paper belongs in the broader area of data augmentations, introducing a method for data augmentation using pretrained diffusion models. The idea is to generate hard negative samples from source samples and contradictory target prompts. A major limitation of this method is the high computational cost, while there seem to be similar approaches to this method in the literature already. The suggestion for this paper is tough, since the reviewers find some merit in the new method, but at the same time the two aforementioned drawbacks are not addressed. Given that there are no theoretical guarantees, nor new technical components, I am going to suggest the authors to increase the novelty before the next iteration.